# Short Communication: The Wasserstein distance as a dissimilarity metric for comparing detrital age spectra, and other geological distributions

Alex Lipp[1] and Pieter Vermeesch[2]

[1]Merton College, University of Oxford, Oxford, UK
[2]Department of Earth Sciences, University College London, London, UK

**Correspondence:** Alex Lipp (alexander.lipp@merton.ox.ac.uk)

**Abstract.**

Distributional data such as detrital age populations or grain size distributions are common in the geological sciences. As analytical techniques become more sophisticated, increasingly large amounts of distributional data are being gathered. These advances require quantitative and objective methods, such as multidimensional scaling (MDS), to analyse large numbers of samples. Crucial to such methods is choosing a sensible measure of dissimilarity between samples. At present, the Kolmogorov-Smirnov (KS) statistic is the most widely used of these dissimilarity measures. However, the KS statistic has some limitations such as high sensitivity to differences between the modes of two distributions, and insensitivity to their tails. Here we propose the Wasserstein-2 distance ($W_2$) as an additional and alternative metric for use in geochronology. Whereas the KS-distance is defined as the maximum vertical distance between two empirical cumulative distribution functions, the $W_2$-distance is a function of the horizontal distances (i.e., age differences) between observations. Using a variety of synthetic and real datasets we explore scenarios where $W_2$ may provide greater geological insight than the KS statistic. We find that in cases where absolute time differences are not relevant (e.g., mixing of known, discrete age peaks), the KS statistic can be more intuitive. However, in scenarios where absolute age differences are important (e.g., temporally/spatially evolving sources, thermochronology, and overcoming laboratory biases) $W_2$ is preferable. The $W_2$-distance has been added to the R package IsoplotR, for immediate use in detrital geochronology and other applications. The $W_2$ distance can be generalised to multiple dimensions, which opens opportunities beyond distributional data.

## 1 Introduction

A distributional dataset is one where the information does not lie in individual observations, but in the *distribution* of many observations associated with one sample. Such data are common in the geological sciences, for example, detrital mineral ages or grain size distributions. Zircon U-Pb ages, in igneous and detrital samples, are one particularly widely used class of distributional data, which are used *inter alia* to constrain sediment provenance, global magmatic processes, and the evolution of plate tectonics (e.g., Condie et al. 2009; Cawood et al. 2012; Reimink et al. 2021). Grainsize distributions are another common form of geological distributional data. Analytical advances mean that increasingly large amounts of distributional

data are being generated in the Earth sciences meaning that qualitative comparison of samples is becoming infeasible, and objective dissimilarity metrics between samples must be used. Some measure of dissimilarity (or more specifically, distance) is also required for many widely used statistical methods such as clustering, ANOVA, and dimension reduction. Dissimilarity metrics in geochronology at present are most commonly used for dimension reducing techniques such as multi-dimensional scaling (MDS) or principal component analysis (PCA). Such methods have become popular for analysing large numbers of detrital age spectra simultaneuously (Vermeesch, 2013; Sharman et al., 2018; Vermeesch, 2018a). Fitting models (e.g., sediment source partitioning) using distributional data also requires a definition of dissimilarity for comparing observed and predicted distributions (e.g., Amidon et al. 2005; De Doncker et al. 2020).

For all uses, the choice of which dissimilarity metric to use is vital as different metrics result in different numerical results and thus different geological interpretations. In general, the most appropriate metric will depend on the data being analysed and the scientific question under investigation. The Kolmogorov-Smirnov (KS) distance, calculated as the maximum vertical distance between two empirical cumulative distribution funtions (ECDFs) has emerged as a 'canonical' distance metric between mineral age distributions (Berry et al., 2001; Vermeesch, 2018a). However, the KS-distance has a number of drawbacks, chiefly that as only the *maximum* vertical difference between ECDFs is important, it is insensitive to variability in the tails of distributions. A number of alternative dissimilarity measures have previously been proposed to address this issue, including established methods such as the Kuiper statistic, and ad-hoc dissimilarity measures such as the 'likeness' and 'cross-correlation' coefficients (Satkoski et al., 2013; Saylor et al., 2012). Unfortunately, these alternatives have drawbacks, including a propensity for the ad-hoc dissimilarity measures to produce unintuitive results when applied to extremely large and/or precise datasets (Vermeesch, 2018a).

In this paper we present an alternative to the KS-distance that does not suffer from some of these limitations: the Wasserstein distance (also known as the Earth-mover's or Kantorovich–Rubinstein distance). To introduce the chief principle behind this measure, let us consider a simple toy example. Table 1 contains four samples ($A$ through $D$), each of which contains exactly one single grain analysis:

**Table 1. A toy, single-grain per sample dataset**

| Sample | A | B | C | D |
|---|---|---|---|---|
| Age, Ma | 1 | 1 | 2 | 11 |

As the KS distance is the vertical difference between ECDFs, it is insensitive to the absolute, 'horizontal' age differences between individual observations. Thus, the KS-distances between $A$ and the other three samples are $KS(A, B) = 0$, $KS(A, C) = 1$ and $KS(A, D) = 1$. Counter to our expectation, the KS-distance cannot 'see' the relative age difference between sample $A$ and samples $C$ and $D$. For the toy example, the Wasserstein distance simply corresponds to the horizontal distance between the four samples. Thus, $W(A, B) = 0$, $W(A, C) = 1$, and $W(A, D) = 10$, which is a more sensible result than that achieved with the KS-distance.

In the following sections, we first introduce the Wasserstein distance in a more realistic setting, and formally define it. Next we discuss how it can be decomposed into intuitive terms that accord with how qualitatively, as geologists, we might compare distributions. We then proceed to compare the Wasserstein distance to the KS distance using a simple yet realistic synthetic example. Finally, we analyse a series of case studies, analysing real datasets using both the Wasserstein and KS distances. We thus evaluate the benefits and drawbacks of both metrics, identifying scenarios when one metric may be preferred to the other. Whilst we focus primarily on detrital age distributions, we emphasise that much of the following discussion applies equally to any form of distributional data.

## 2 The Wasserstein distance

The Wasserstein distance is a distance metric between two probability measures from a branch of mathematics called 'optimal transport'. Optimal transport is often intuited in terms of moving piles of sand from one location to another with no loss or gain of material (e.g., Villani 2003). The problem that optimal transport solves is finding the way to transport the sand such that the least sand is moved the least distance. The Wasserstein distance is the cost associated with this most efficient transportation. The association with moving piles of sand is why the Wasserstein distance is often termed the Earth-mover's distance. Figure 1a shows an example of how one univariate probability distribution, $\mu$, based on a detrital age spectrum, is transformed into another, $\nu$ according to the optimal transport plan. Elsewhere in the Earth sciences, the Wasserstein distance is increasingly used for solving non-linear geophysical inverse problems (e.g., Engquist and Froese 2014; Métivier et al. 2016; Sambridge et al. 2022) and has been proposed as a tool for fitting hydrographs (Magyar and Sambridge, 2023). Full mathematical treatments of the Wasserstein distance and optimal transport are beyond the scope of this paper, but interested readers are referred to Villani (2003) or Peyré and Cuturi (2019). A geophysical perspective is given in Sambridge et al. (2022).

### 2.1 Formal definition

We consider two univariate probability distributions $\mu$ and $\nu$ which have cumulative distribution functions (CDFs) $M$ and $N$ respectively. The $p^{\text{th}}$ Wasserstein distance between $\mu$ and $\nu$ is given by:

$$W_p(\mu,\nu) = \left( \int_0^1 |M^{-1} - N^{-1}|^p \mathrm{d}t \right)^{1/p}.$$ (1)

where $M^{-1}$ indicates the inverse of the CDF $M$ and $0 \leq t \leq 1$ (Villani, 2003). Note that this definition of $W_p$ assumes that the cost-function is given by $|x - y|^p$ (e.g., the Euclidean distance where $p = 2$), which is the case for most distributional data in geology. In the further special case of $p = 1$ (i.e., the *first* Wasserstein distance, $W_1$), Equation 1 can be re-written simply as:

$$W_1(\mu,\nu) = \int_X |M - N| \mathrm{d}x,$$ (2)

which is the area between two CDFs (e.g., Figure 1b). Recall that the KS-distance between two distributions is the maximum distance between the two corresponding CDFs. Whilst the $W_1$ is easily visualised, we actually use the $W_2$ going forwards as

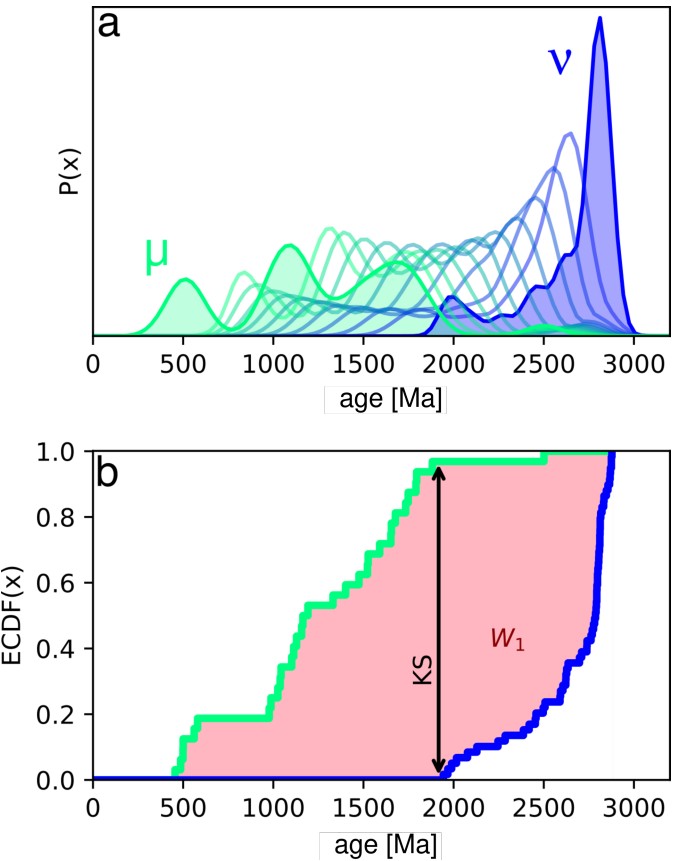

**Figure 1. Intuition of the Wasserstein distance.** a) Green and blue filled polygons show two example kernel density estimates of mineral ages from two samples (based on data from Morton et al. 2008) The distributions are labelled $\mu$ and $\nu$ for consistency with Equation 1. Semi-transparent coloured lines are probability distributions spaced equally in Wasserstein space between $\mu$ and $\nu$ (termed 'barycentres'; Benamou et al. 2015). b) Empirical Cumulative Distribution Functions (ECDFs) of the detrital ages used to calculate the distributions shown in panel a, same colours. The first Wasserstein ($W_1$) distance corresponds to the total area between the two ECDFs (shaded pink). The Kolmogorov-Smirnov (KS) distance is the maximum distance between the two ECDFs (black double-headed arrow).

the *squared* distance (i.e., $p = 2$) between observations is the standard distance metric in most statistical analyses (e.g., least squares regression). Additionally, $W_2$ decomposes into readily interpretable terms, as discussed below.

We focus on these univariate instances as they apply to the most common geological distributional data including detrital age distributions and grain size distributions. However, we note that the Wasserstein distance is, in general, multivariate. As a result, some form of the Wasserstein distance could prove useful for analysing a number of other geological datasets such as the geochemical compositions of detrital minerals, or joint U-Pb and Lu-Hf isotope analysis (see Vermeesch et al. 2023). Statistics for comparing distributional data in multiple dimensions are increasingly needed (Sundell and Saylor, 2021).

Like the KS distance, $W_2$ satisfies the triangle inequality, and as such is a true metric. This property means that classical, as well as metric & non-metric MDS can be used with a $W_2$ defined dissimilarity matrix. As $W_2$ is sensitive to absolute time differences, metric (or classical) MDS, which seek to preserve absolute distances, may be preferable to non-metric MDS. For the rest of this manuscript, metric MDS is used.

## 2.2 Decomposition

A particularly useful property of $W_2$ between two univariate distributions is that it can be decomposed in terms of the differences between the two distributions' location, spread and shape. Irpino and Romano (2007) show that:

$$W_2^2(\mu, \nu) = \overbrace{(\bar{\mu} - \bar{\nu})^2}^{Location} + \overbrace{(\sigma_\mu - \sigma_\nu)^2}^{Spread} + \overbrace{2\sigma_\mu\sigma_\nu(1 - \rho^{\mu\nu})}^{Shape}, \tag{3}$$

where $\bar{\mu}$ is the mean of $\mu$, $\sigma_\mu$ is the standard deviation of $\mu$ and $\rho^{\mu\nu}$ is the Pearson correlation coefficient between the quantiles of the distributions $\mu$ and $\nu$. These three terms also accord with, qualitatively, how as geologists we might compare two distributions.

## 2.3 Discrete data

Most distributional data in the Earth sciences do not, in raw form, follow continuous probability distributions. Instead, samples may be discrete sets of observations, e.g., lists of individual mineral ages. The above formulations can be easily applied to such cases by describing the probability functions $\mu$ and $\nu$ as weighted sums of $\delta$ functions. For example, let us consider two samples $x_m$ and $x_n$ with $p$ and $q$ numbers of observations respectively:

$$\mu = \sum_i^p m_i \delta_{x_m}, \quad \nu = \sum_i^q n_i \delta_{x_n} \tag{4}$$

where $m$ and $n$ are weight vectors, such that $\sum m_i = \sum n_i = 1$. In most geological cases these weights would be uniform, $m_i = 1/p$; $n_i = 1/q$, giving each observation within a sample equal weight. In this scenario, $M$ and $N$ are the familiar empirical cumulative distribution functions (ECDF), given as a series of step functions (e.g., Figure 1b).

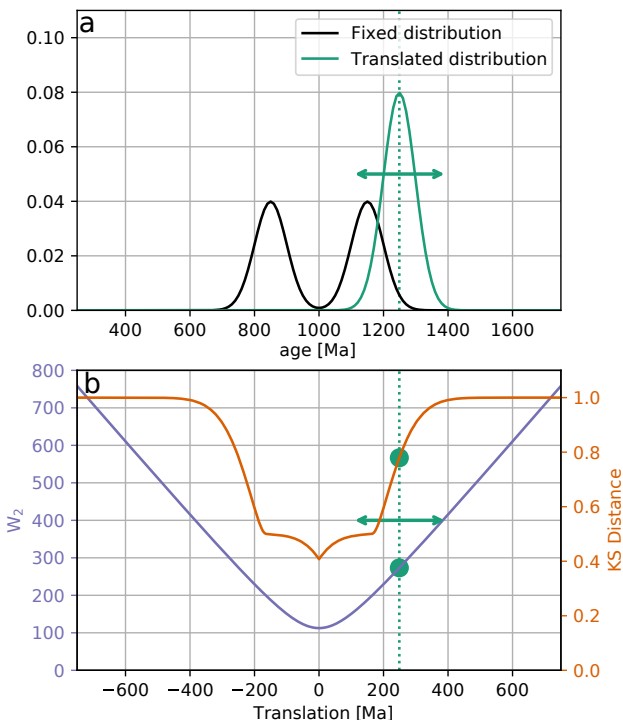

**Figure 2. Comparing the Wasserstein distance to the Kolmogorov-Smirnov distance.** a) Two synthetic probability density functions, modelled on U-Pb age spectra. The black bimodal distribution is fixed at 1000 Ma, and the green unimodal distribution is translated along the time axis. b) For each translated distribution, we calculate the KS-distance (red line) and $W_2$ (blue line). The green dashed line and circles indicate values associated with the location of the green distribution shown in panel a.

### 2.4 A synthetic example

To demonstrate the intuition of $W_2$ we explore a simple synthetic example. We consider two probability density functions of mineral ages: a bimodal distribution and a unimodal distribution, both constructed from Gaussians with the same scale (Figure 2a). We fix the bimodal distribution at 1000 Ma, but translate the unimodal distribution along the time axis. For each translated distribution we calculate both the KS-distance and $W_2$. Figure 2b displays the behaviour of both distances under this scenario. The KS-distance shows an unexpectedly complex response containing a series of steps, as the peaks of the distributions align and misalign. At around $\pm 400$ Ma, once the distributions stop overlapping, the KS-distance plateaus at its maximum value of 1. By contrast, $W_2$ increases monotonically with increasing distance. Away from the origin, $W_2$ approximates a linear function of the amount of translation, as is predicted from Equation 3. At the origin, the non-zero value of $W_2$ is the cost of turning the unimodal distribution into the bimodal distribution without translation.

We argue that the behaviour of $W_2$ is more geologically intuitive than the KS-distance under this scenario. It is useful geological information if two distributions differ in their means by 400, 500 or 1000 Ma, but if the distributions do not overlap, the KS-distance is insensitive to this. The Wasserstein distance is, by contrast, sensitive to the absolute offset between non-

overlapping distributions. Additionally, the stepped response of the KS-distance under translation is undesirable. Under the simple operation of translating a unimodal distribution, we would expect our dissimilarity to increase at a constant, or at least predictable (e.g., quadratic) rate. The change of the KS-distance with translation is, unintuitively, non-linear. By contrast, the $W_2$ increases linearly with respect to translation.

We reiterate that at a translation of 0 Ma, $W_2$ (and the KS distance) is still non-zero, reflecting the fact that even when the average ages are aligned, the shapes of the uni-modal and bi-modal distributions do not match. This illustrates the tendency of $W_2$ in geochronological data to prioritise aligning the average ages of distributions *before* considering matching individual peaks. Such behaviour contrasts with approaches that seek to only match probability peaks neglecting any information of absolute ages (e.g., Saylor and Sundell 2016).

## 3    Discussion

As stated above, the most appropriate dissimilarity metric to use will depend on the scientific question being answered. In general, the Wasserstein distance is most appropriate when absolute differences along the time axis (or more generally, the x-axis) provide useful information to solving the geologic problem. The KS distance however is more appropriate when the size of the time differences between peaks is not relevant. Both the KS distance and the $W_2$ are calculated in terms of differences between ECDFs. Due to these similarities in construction, in many cases the results from using the KS and $W_2$ are, encouragingly, similar. One exception is whether ages are log transformed prior to analysis. Because the KS distance considers only the order of the ages, it will be the same whether a log transform is used or not. $W_2$ however will be different, and will consider *relative* not absolute age differences. Such an example is discussed below (Figure 5).

Here we discuss a variety of realistic scenarios where the KS and $W_2$ may result in different interpretations. In each, we evaluate the advantages and disadvantages of using $W_2$ or KS. These case-studies can be used to determine which metric is most appropriate for a particular scenario.

### 3.1    Discriminating contributions from discrete endmembers

We first consider a scenario where the samples are assumed to be mixtures, in differing proportions, of some known or unknown fixed endmembers. This situation is one where absolute distance along the time-axis is not relevant, as the nature of the endmembers is not sought, simply their relative contributions to a set of mixtures. Instead, it is *vertical* differences in the probability at a given age that is relevant. The KS distance, which is sensitive to such vertical differences in age distributions is better suited for this than $W_2$. Indeed, in such a scenario the $W_2$ can result in some unintuitive behaviour.

For example, let us consider three unimodal potential sediment sources, as shown in Figure 3a. We now consider two mixture samples. The first is an equal mixture of X and Y, and the second an equal mixture of Y and Z (bottom two plots, Figure 3a). Geologically, we would expect these samples to be about half as similar to the two source endmembers. However, a $W_2$ MDS map identifies these samples as being removed from their two endmembers (Figure 3b). Additionally, because of the absolute time difference between Source Z and the other sources, Sample 2 is treated as a considerable outlier. The KS distance performs

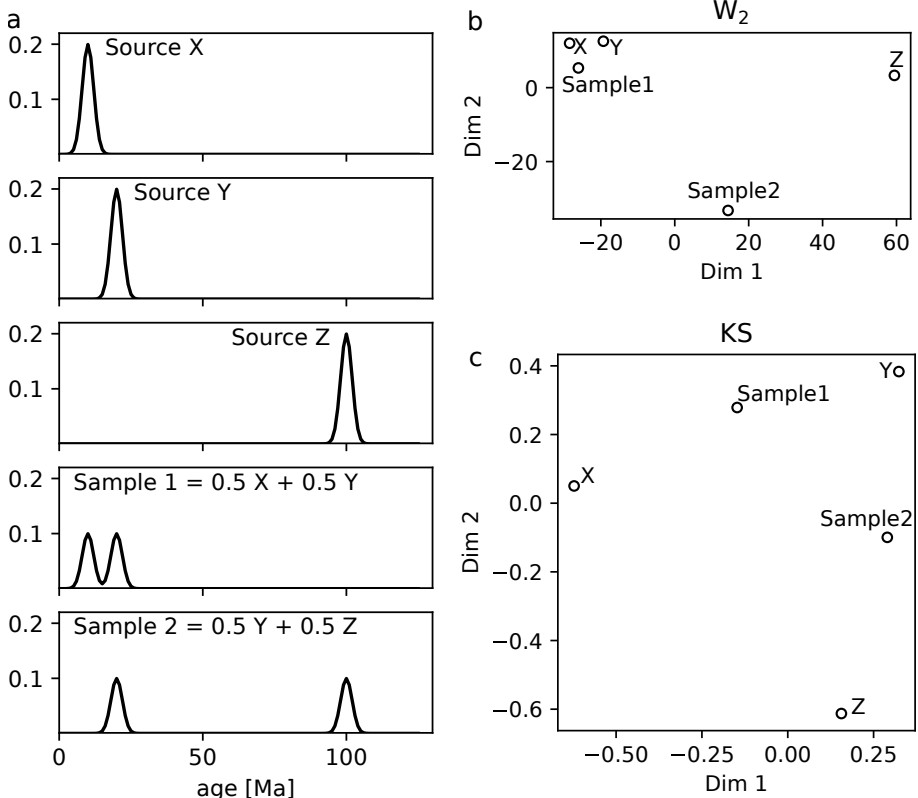

**Figure 3. Mixing of discrete endmembers** a) Three theoretical, unimodal source age distributions with peaks at 10, 20 and 100 Ma, and two mixture samples. Sample 1 is an equal mixture of X and Y and Sample 2 a mixture of Y and Z. b) Metric MDS map of the three sources and the mixtures using $W_2$ distance (stress = 0.05). c) Same as panel b for KS distance (stress = 0.05). This is a scenario where KS distance may be preferable to $W_2$.

better here, placing the mixtures approximately halfway between the expected endmembers. However, in such a well defined mixing scenario as this, methods such as endmember mixture modelling may be more appropriate than statistical dimension reduction (e.g., Weltje 1997; Sharman and Johnstone 2017; Dietze and Dietze 2019).

## 3.2 Temporally varying source age distributions

In contrast, scenarios where the shape of sediment source age distributions evolves in space and time are well suited to using $W_2$. This is because $W_2$ considers all parts of a distribution, whereas the KS only compares one point, the location of maximum ECDF separation. For example, Figure 4 displays detrital zircon age distributions gathered by DeGraaff-Surpless et al. (2002) from sediments from a section (Cache Creek) across the Great Valley Group in California, USA. The age populations are shown as KDEs and histograms, in stratigraphic order, in Figure 4a. The uppermost samples show an increasingly broad distribution

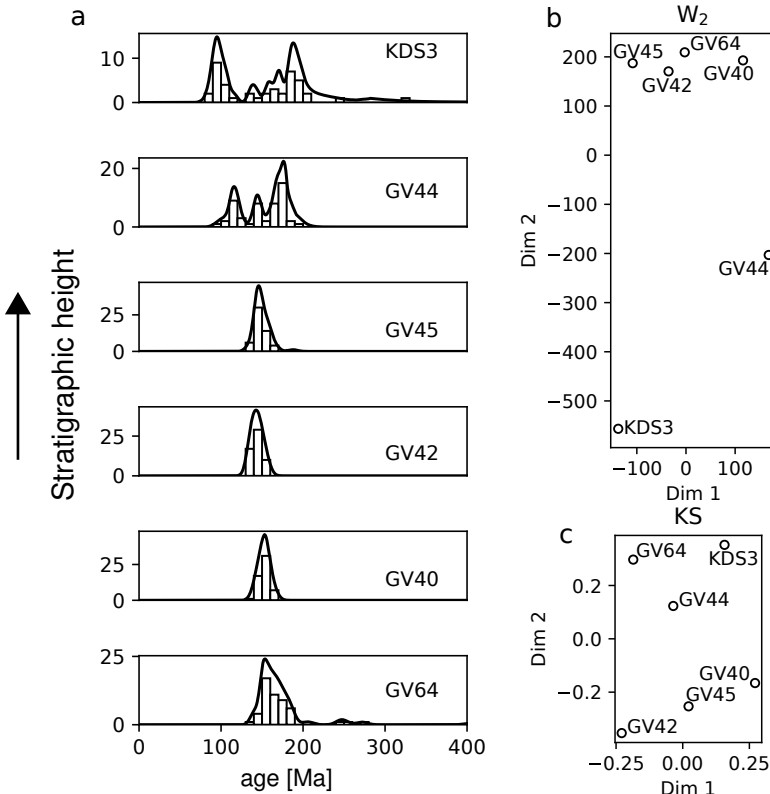

**Figure 4. Temporally evolving source distributions.** a) KDEs and histograms for zircon age distributions for samples from Cache Creek section across Great Valley Group, arranged in stratigraphic order (DeGraaff-Surpless et al., 2002). b) MDS map using $W_2$ for data shown in panel a (Stress = 0.28). c) Same as b using KS distance (Stress = 0.18). In this scenario, the results from $W_2$ are preferable.

than the lower four unimodal samples. DeGraaff-Surpless et al. (2002) attribute this trend, *inter alia*, to expanding sediment source areas.

Figures 4b–c display MDS maps calculated using $W_2$ and KS respectively. The $W_2$ map clearly identifies the stratigraphic order of the samples by the changing distribution shape. Additionally, it clusters the four unimodal samples together. By contrast, the KS map does not identify the stratigraphic trend, locating the lowermost stratigraphic sample GV64 with the uppermost samples KDS3 and GV44. We conclude then that the $W_2$ has better captured the geological information in this scenario.

**3.3   Thermochronology**

In thermochronology, age distributions shift along the time-axis according to thermal signals (e.g., exhumation). In many thermochronological studies, we may seek to characterise how such a signal evolves in space and time. For this question

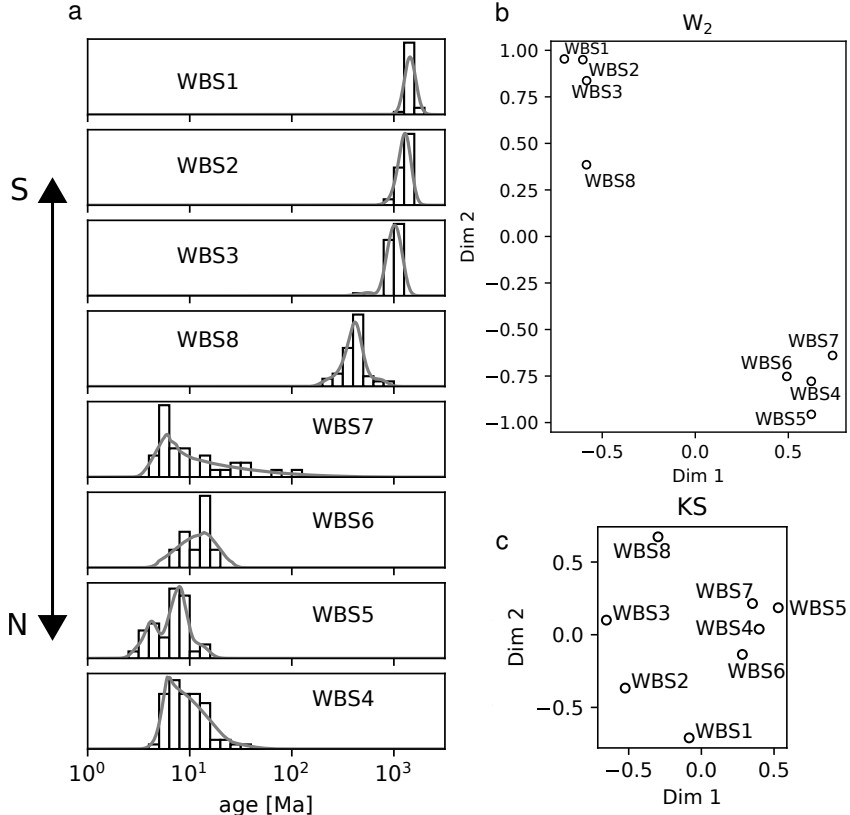

**Figure 5. Analysing thermochronological data using $W_2$ and KS distances.** a) KDEs for a detrital mica $^{40}$Ar/$^{39}$Ar dataset of Wobus et al. (2003) arranged from south to north across a physiographic transition of the central Himalaya in Nepal. Note the logarithmic scale. b) The MDS configuration using $W_2$, following a log transform (stress = 0.02). c) MDS map using KS statistic (stress = 0.18). In this example, $W_2$ performs better than the KS distance at identifying the geographic trend.

absolute distance along the time-axis is useful information and so the $W_2$ may be more effective than the KS distance. For example, Wobus et al. (2003) use $^{40}$Ar/$^{39}$Ar detrital mica thermochronometry to explore spatially varying exhumation along

a spatial transect in the Himalaya. The KDEs of the samples are shown in Figure 5a arranged south to north. The southern samples (WBS1, WBS2, WBS3, WBS8) show old exhumation signals, but a dramatic shift to younger ages is observed north of a distinct physiographic transition. MDS maps of these samples are shown using the KS distance and $W_2$ in Figures 5b–c respectively. As there is limited overlap between the samples, the KS distance struggles to capture the NS progression in exhumation age. Whilst the physiographic division is found, it weights it equally to variation within one cluster. By contrast,

the $W_2$ map correctly identifies the simple temporal and geographical trend of the samples from south to north.

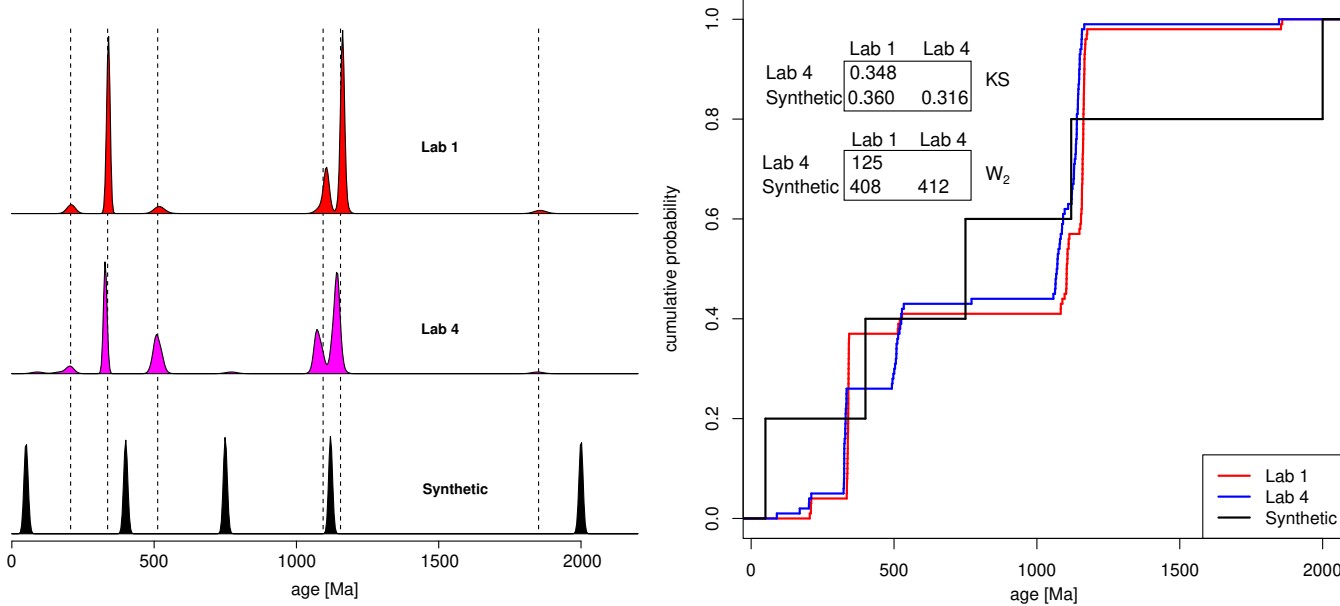

**Figure 6. Comparing samples from an inter-laboratory calibration study**. KDEs (left) and ECDFs (right) of two samples from the inter-laboratory comparison study of (Košler et al., 2013), plus a purposefully misaligned synthetic sample. Dashed lines mark the true ages of the detrital mixture. According to the KS-statistic, the age distribution produced by Lab 4 is more similar to the synthetic distribution than it is to the distribution produced by Lab 1, despite the absence of any shared age components. The $W_2$ distance correctly deems the distribution produced by Lab 4 to be closer to that of Lab 1 than to the synthetic mixture.

## 3.4 Combining data from multiple laboratories

A final scenario where the $W_2$ could be preferable is when comparing samples from different laboratories which are affected by inter-laboratory bias. Košler et al. (2013) provided ten different laboratories with identical synthetic zircon samples with a known age distribution. Different instruments introduced small differences in the ages of each peak. For example, in Figure 6

we display the results from Lab 1 (red) and Lab 4 (pink) as KDEs. The expected peak at $\sim 1200$ Ma (dashed line) is offset between the two samples. As it is the maximum distance between two ECDFs, the KS distance is very sensitive to minor offsets in sharply defined peaks. In this case, the KS distance between these theoretically identical samples is large at 0.348, which is over one third of the maximum possible distance between samples. Indeed, the KS distance considers a synthetic, purposefully misaligned series of peaks (black KDE) to be more similar to the Lab 4 results than the results from Lab 1. The $W_2$ distance,

does not suffer from this oversensitivity to minorly offset peaks and correctly identifies the samples from Lab 1 and Lab 4 as being much more similar than the random synthetic distribution.

## 4 Implementation

We provide example code (`github.com/AlexLipp/detrital-wasserstein`) in both Python and R that demonstrates how to calculate the $W_2$ between two univariate distributions (U-Pb zircon ages). For these examples we make use of the the POT and transport packages in Python and R respectively which implement solutions to Equation 1 (Flamary et al., 2021; Schuhmacher et al., 2022).

### 4.1 IsoplotR

Additionally, the $W_2$-distance has been added to the IsoplotR package in R, which calculates dissimilarity matrices and MDS maps (Vermeesch, 2018b). This software can be accessed using an (online) graphical user interface, at `isoplotr.es.ucl.ac.uk`. Alternatively, the function can also be accessed from the R command line. The following snippet uses $W_2$ to calculate an MDS map for the dataset from Wobus et al. (2003) discussed in the manuscript (Figure 5). The data required is also available at the above repository. Note that the MDS map produced may show slight differences to those in the manuscript due to dependence of metric MDS on a random state variable. This variability can introduce reflections/rotations of the data, but the underlying structure is unchanged.

```
# load the package:
library(IsoplotR)

# Load in the data
DZ <- read.data("wobus.csv",method="detritals")

# example 1. calculate the W2 distance matrix for the dataset:
d <- diss(DZ,method="W2")

# example 2. apply MDS to the dataset:
mds(DZ,method="W2")
```

## 5 Conclusions

The second Wasserstein distance, $W_2$, is an effective metric for comparing distributional data in the geological sciences such as detrital age spectra or grain size. Unlike the KS distance, $W_2$ can be extended to further dimensions. $W_2$ is a function of the horizontal distances between observations, in contrast to the KS distance, which corresponds to vertical differences between ECDFs. Using a variety of case studies we explore scenarios where the $W_2$ may or may not be preferable to the KS distance. In scenarios where discrete, known age peaks are mixed, the KS distance may be preferable. However, in other scenarios where absolute differences along the time axis are useful information, $W_2$ is preferable. Example scenarios include

spatially/temporally evolving source distributions, thermochronological data, and combining detrital samples from different laboratories. The Wasserstein distance has been added to the IsoplotR software, and example scripts are provided in Python

and R.

*Code availability.* The code and data repository is found at `github.com/AlexLipp/detrital-wasserstein`

*Author contributions.* AGL conceived the project, both authors contributed to development, writing, and software production.

*Competing interests.* PV is an Associate Editor of Geochronology

*Acknowledgements.* AGL is funded by a Junior Research Fellowship from Merton College, Oxford. PV is supported by NERC Standard
Grant #NE/T001518/1. This work benefited from discussions with Malcolm Sambridge & Kerry Gallagher. We thank reviews from Joel Saylor, an anonymous reviewer, and the associate editor Michael Dietze for their constructive feedback.

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
