# Peer review of "Short Communication: The Wasserstein distance as a dissimilarity metric for comparing detrital age spectra, and other geological distributions"

_EGUsphere, 2022_

## Referee Comment (RC1)

**Referee contribution to public discussion - Geochronology**

invitation received: 2022-11-18 | today: 2022-11-22

**1    Contribution summary**

In a nutshell, the manuscript presents a numerical feature implementation in R and a code example in Python to map similarities/dissimilarities between distributional age data. The 'new' metric is the Wasserstein-2 distance, which is somewhat tested against the Kolmogorov-Smirnov distance metric.

**2    Major comments**

**2.1    Presentation quality**

Sincerely, I enjoyed reading the manuscript. It is a concise and neatly written technical paper that presents reasoning and implementation of a numerical metric to map age/grains-size distributions in R and Python. The graphical quality of the figures and tables is good. Still, the axes labelling could be improved for readers unfamiliar with multidimensional scaling.

Minor point: The title seems to promise more than the manuscript delivers. "Comparing . . . ". The manuscript does not compare something. The presented "comparison" is a performance test of the Wasserstein-2 distance and the Kolmogorov-Smirnov distance. I think the title should reflect better what the manuscript tries to achieve: a presentation of an alternative metric in the realm of multidimensional scaling.

**2.2    Scientific significance**

The general idea of the manuscript fits within the scope of GChron. However, I am in a little bit of doubt about whether it justifies requesting a peer-review procedure and a peer-review publication. The numerical metric presented here is not new, and the manuscript does not (yet?) show significant scientific progress. The implementation in R appears limited to a few code lines. Perhaps under different circumstances, the implementation in R would have remained a single line in a news files, along with a few lines in the package manual or an entry in a science blog.

Having said that, on the other hand, I understand that the authors of such software solutions usually spend countless hours on coding and maintaining valuable scientific code used over and over for free by others. At some point, they must cash in their effort to get the credit they deserve. Nonetheless, I believe the manuscript content is not significant enough, and the authors should put a little more effort into it. For instance:

- As a non-expert in multidimensional scaling, I feel the manuscript would benefit from more context. The formal description is sufficient and easy to follow, but the likely impact of this manuscript seems low except for having announced a 'new' feature. In other words: How does this new measure perform for real samples and their (new) interpretation? Section 4 reads interesting, but was a new conclusion reached? Did it lead to better (e.g., more accurate, more precise) results, or did the geoscientific interpretation essentially remains the same? If the latter is the case, perhaps you can present a real case underlining the point you want to make better.

- The manuscript comes without a proper discussion. Section 4 is an application example that includes elements of a discussion. However, for a scientific manuscript, I would expect to see more. In particular I would like to see a discussion about the question: Does it likely change the outcome of studies working with this 'new' metric.

- The synthetic data outlines the general problem you want to address. I suggest leading with an example based on a case study where the Kolmogorov-Smirnov distance did not perform as expected for the reasons you have mentioned.

**2.3 Scientific quality**

The scientific quality of the manuscript is good and valid. I found a few minor inconsistencies though, but nothing out of the ordinary (see detailed comments below).

**3 Detailed comments**

- L111: I've played a bit with the proposed synthetic data and found that it depends to some extent on the standard deviation. A more narrow standard deviation for the same fixed mean values leads to more complex KS-distance patterns. The higher the degree of overlap (higher standard deviation), the more conclusive the KS distance becomes. Perhaps you can add a few lines about it in the text.

- L150-L175: I think this paragraph can be improved in order to provide a better experience to readers.

  - I had to download the example files from an external repository, but I was expecting to find everything up and running as a supplement to the manuscript; a minor issue, though.

  - I was expecting the R and the Python code snippets to do somewhat the same; just for the two different languages. Instead, the R code loads a CSV file with eight datasets, and the Python code imports only two datasets. The R code does considerably more. It will be easier to understand if the example code lines lead to the same output (they do if I limit the R code example to the datasets of the Python code).

  - The R code snippet produces a plot output. However, if I reduce the dataset, it fails. This appears to be a bug in the package 'IsoplotR' because it returns an uncontrolled error:

```r
DZ <- IsoplotR::read.data("scandinavia_short.csv", method = "detritals")
DZ

**$Byskealven**
**[1] 1507 1769 1762 1077 1246  943 1776 1453 1129 1875 1847 1792 1286 1870 1798**
**[16] 1811 1806 1016 1590 1798 1834 1794  989 1457 1832 1856 1794 1787 1809 1875**
**[31] 1790 1816 1740 1878 1698 1739 1593 1811 1803 1868 1795 1710 1805 1419 1635**
**[46] 1800 1606 1622 1865 1813  968 1627 1497 1812 1782 1823 1813 1387 1623**
##
**$Vefsna**
**[1] 1677 1113  987 2501  500 1655  977 1748 1526 1401 1882 1025 1192 1129 1164**
**[16] 1522 1652 1734 1157  499 1795  486 1475 1791 1331  580 1043  455 1590 1102**
**[31] 1038  561**
##
**attr(,"class")**
**[1] "detritals"**
```

```r
**example 1. calculate the W2 distance matrix for the Scandinavian dataset:**
d <- IsoplotR::diss(DZ, method = "W2")

**[1] "W2"**

d

**Byskealven**
**Vefsna      490.0072**
```

```r
**example 2. apply MDS to the Scandinavian data set:**
try(IsoplotR::mds(DZ, method = "W2"))
```

```
**[1] "W2"**
**Error in cmdscale(d, k) : 'k' must be in {1, 2, ..  n - 1}**
```

- – I would not call the Python code "implementation", which, to me, implies new, unique software code. Instead, the Python code is a minimum running example showing how $W_2$ values can be calculated in Python using existing libraries.

- – It is probably evident to the authors that both attempts, R and Python, lead to similar results, but this should be made clear to the readers by quoting the output for each code snippet. Or, if this is too obvious, remove the output after the Python code.

- – L152: If I look into the R code (file `mds.R`), I read in line 199 of the code: `#modified after the wasserstein1d function of the transport package`. It is normal to look up open-source code of others, however, if it helped for the own implementation and since the code line in question seems identical, credit should be given in the manuscript to authors of the package `'transport'` (Schuhmacher et al. 2022)

```
**transport file transport1d.R, line 6**
return(mean(abs(sort(b)-sort(a))^p)^(1/p))

**Isoplot, file mds.R, line 199**
out <- mean(abs(sort(y)-sort(x))^p)^(1/p)
```

- L164: Please consider adding the example data to the manuscript or the R package

- L167+ (footnote): The repository `pvermees/IsoplotRbeta` does not exist, but I guess the branch `beta` was meant and it should read: `remotes::install_github("pvermees/IsoplotR@beta")`

**3.1 Comments on figures and tables**

- Figure 1:

  - – I suggest using dashed lines for the "semi-transparent colour lines" for better readability.

  - – How did you modify the data to "aid illustration"? It appears that you have shifted the 'Byskealven' dataset by 1 Ma. This should be mentioned. If so, how does it affect the $W_1$ distance? Your pink area becomes considerably smaller if the non-shifted dataset is used. Perhaps you have a better dataset at hand; one that does not need such manipulation.

- Figure 2:

  - – The upper plot would benefit from y-axis labelling

- Figure 3:

  - – Similar to my comments above, please label the plot axes. I do not doubt that readers familiar with this kind of analysis will have no problem understanding what is shown, but please consider the GChron's broader audience.

  - – Something is at odds with the lower figures (j and k), if I try to reproduce them with `IsoplotR::mds()` and the example data from GitHub. Figure 3j does not look as in the manuscript, because 'Ljusnan' and ' Byskealven' seem to be no different. Figure 3k is somewhat mirrored. Below my code, I first used `IsoplotR():mds()`, here more manually to show the calculation steps. The mirrored figure is not a big deal because the interpretation should not change, but it should be presented as the users would see it running the code.

```
DZ <- IsoplotR::read.data("scandinavia.csv", method = "detritals")
**calculate xy values for plots**
d_W2 <- IsoplotR::diss(
  x = DZ,
  method = "W2") |> MASS::isoMDS(trace = FALSE)
```

```
**[1] "W2"**
d_KS <- IsoplotR::diss(
  x = DZ,
  method = "KS") |> MASS::isoMDS(trace = FALSE)
```

```
**[1] "KS"**
```

- I am not sure how this table helps the readers. The information given is of no particular relevance to the text. Either extend the table and add additional information or remove the table. Later I saw in the text why the 2nd column was in the table. I still think you could add more details. Otherwise, it appears relatively trivial.

**4 Additional comments**

The following comments refer to something I discovered along the way. It was not considered for my recommendation to the editor.

- I have not used `'IsoplotR'` before, but it appears to be an interesting and mature package. However, something I noticed missing while searching for the new feature implementation was a NEWS file, as it is custom for R packages. The authors may want to consider adding such a file because it helps users understand package changes without inspecting code changes. This is something I find very useful for scientific software packages.
- The R code example returns a default plot (`IsoplotR::mds()`). Perhaps it is obvious to users familiar with the package, but I found it confusing having numbers on the x and y axis but no axes labels. I guess what is shown is some scaled distance. In Section 4, you have explained it better, but personally, I prefer to understand figures without reading the text.
- The default plot `IsoplotR::mds()`returns only the names, but the location in the plot remains a little bit obscure because the default plot setting is `pch = NA`. I find it awkward because it reduces the value of the plot when what is shown remains too vague to be positioned correctly. In particular, because the `text()` position in R can be very different from what you would expect depending on the setting in `pos`.
- In the function `IsoplotR::plot.MDS(),` the standard plot output has the argument `asp = 1` hard coded. This is at least unexpected and leads to inconclusive plot behaviour' when `ylim` and `xlim` are modified.

**References**

Schuhmacher, Dominic, Björn Bähre, Carsten Gottschlich, Valentin Hartmann, Florian Heinemann, and Bernhard Schmitzer. 2022. *transport: Computation of Optimal Transport Plans and Wasserstein Distances.* https://cran.r-project.org/package=transport.

---

## Referee Comment (RC2)

**Summary**

The authors introduce a new metric to the world of detrital geochronology: the Wasserstein Distance. The authors assert that this metric overcomes shortcomings of the KS distance. The authors define the metric and compare its use to the KS distance in a simple toy example. They then apply the WS Distance and the KS Distance to compare a number of river samples from Scandinavia. They conclude that the WS Distance produces more geologically meaningful comparisons than the KS Distance.

**Recommendation**

The authors have not demonstrated that the WS Distance produces geologically meaningful results. At the very least they need to address the concerns raised below before publication. However, the comments below suggest that the WS Distance may not be an appropriate metric to compare geochronological distributions, because of the unique geological implications of minorly distinct age modes (distinct sources) versus multimodal distributions which include some shared age modes (potentially shared sources).

**General comments**

The WS Distance may be more intuitive than the KS distance in many cases, but whether it is more "sensible" depends on the application. While the toy dataset clearly shows the advantage of WS over KS for assessing simple dissimilarity between sample ages, in many DZ studies it is the degree to which samples share the same sources that is of interest; the absolute difference in age is not directly relevant. For example, if we assume the sources for the samples in the toy dataset each have distinct ages, A and C are no more similar in terms of their sources than A and D, and the equal KS value of 1 (i.e., complete dissimilarity) for (A, C) and (A, D) is actually more informative than the WS values $W(A, C) = 1$ and $W(A, D) = 10$.

The authors need to explore the behavior of WS Distance for mult-modal data sets. For example, what is the interpretation of high versus low WS Distances when comparing multi-modal data sets? This may seem intuitive, however, I suspect that the results will not be intuitive given how the WS Distance is calculated.

The geological context of the samples needs to be provided. Even a reproduction of the Morton et al. (2008) Figure 1 would be helpful in terms of understanding which samples would be predicted to be derived from similar sources.

I am fairly surprised that the authors chose samples with n<=60 grains to showcase a new statistical comparison metric. Multiple studies (including one by the co-author) have shown that n>>100 (ideally n>300) are needed for robust statistical comparison. Although it can be argued that samples of that size are unnecessary for simple age distributions such as those presented here, this raises the question of why choose the simplest possible scenario to showcase a new application of a statistical metric. Rather, it seems that a true breakthrough would be demonstrating that the WS Distance can deal with a previously unresolved problem or elegantly deals with an intensely complex data set.

Figure 2 demonstrates problems with both the KS and WS distances. First, the WS distance increases linearly with displacement away from 1000 Ma. However, once the two distributions no longer overlap, they are no longer any more or less similar because the x-axis is age, not distance.

Two age distributions that share no age modes are equally dissimilar regardless of the age difference between their modes due to the geological implications of sharing versus not sharing age modes. Hence, the increasing WS Distance with displacement beyond zero overlap is an undesirable trait. Second, both the WS and KS distances indicate that the distributions are most alike (minimum WS and KS distance) when both are centered at 1000 Ma. However, at that point they share no age modes. The green distribution would have an age mode at 1000 Ma, but the black distribution would have age modes at ~900 and ~1100 Ma. Rather, they should be most alike when the green distribution overlaps with one of the age modes of the black distribution. As an aside, this is the behaviour that cross-correlation of KDEs of these distributions provides (below).

[Figure]

Figure 3 seems problematic for application of the WS distance to detrital geochronology. For example, Ranealven, Lainioalven, and Bysealven all share peaks at ~1800 Ma. The KS distance correctly locates these samples closest to each other. The WS distance in contrast locates the Ljusnan closer to the Ranealven than the Lainioalven even though the major age modes between the Ranealven and Lainioalven (~100 Myr offset) are closer than those of the Ranealvan and Ljusnan (~200 Myr offset). The WS Distance also locates the Vindelalven and Lainioalven equidistant from the Ranealven. The problem is that distinct source areas may have similar but non-overlapping age modes, and the WS distance is insensitive to these minor differences and therefore unable to discriminate them as distinct sources. A detrital geochronology distribution that has a mode at 1800 Ma and 2800 Ma is more likely to share a source with a sample with a distribution at 1800 Ma (i.e., one overlapping age mode) than one at 1700–1750 Ma (i.e., no overlapping age modes). The authors may be able to address this concern, but it seems to me to be a fatal flaw in this metric.

The problems with this metric can be seen when comparing some more complex age distributions. Simple visual inspection indicates that Samples 1, 2, and 7 should be quite similar.

[Figure]

[Figure]

[Figure]

This is confirmed by a KS-based MDS (above left), or Kuiper-based MDS (above right).

[Figure]

It is also confirmed by a cross-correlation-based MDS (above).

[Figure]

However, a WS Distance-based MDS (above) non-intuitively locates samples 1 and 6 quite close, and locates 1, 6, and 7 closer to 5 than to 2.

A second example highlights the disproportionate impact of misalignment of detrital age modes on the WS Distance. P1 and P3 below share age modes at 1800 Ma. P3 has an additional mode at 500 Ma. P2 shares no age modes with either P1 or P3 and rather has an age mode at 1700 Ma.

[Figure]

This is reflected in their KS MDS plot (above left) and an MDS plot based on cross-correlation (above right, note difference in x- and y-axis scale).

[Figure]

However, the WS Distance MDS (above) does not reflect the true relationship between age modes, but rather reflects the anomalous area between P1 and P3 (i.e., the horizontal distance between 500 Ma and 1800 Ma in the ECDF plot).

**Detailed comments**

The authors assert that cross-correlation is an ad hoc method, yet the Pearson coefficient which is the basis for the cross-correlation coefficient is widely used in seismic analysis, waveform analysis, and image analysis. As such, it is unclear what is meant by "ad hoc" in this context. Although Vermeesch (2018) correctly points out limitations of cross-correlation as applied to probability density plots of extremely precise data, these caveats do not apply to its application to kernel density estimates or kernel functional estimates. Similarly, the charge that Likeness is ad hoc is unfounded. Likeness is an adaptation of the $L^1$ norm applied, usually, to a 1D geochronology distribution. (However, see Sundell et al. (2021) for application of the $L^1$ norm to 2D distributions.) Finally, like the cross-correlation coefficient and Likeness, the Sircombe-Hazelton distance ($L^2$ norm, Sircombe and Hazelton, 2004; Vermeesch, 2018) also requires discretization of continuous functions for its calculation. I guess the take away from this is that it may be useful to compare the performance of cross-correlation and Likeness in addition to the KS Distance to newly applied metrics like the WS Distance.

It is ironic that after railing against use of the cross-correlation coefficient, the authors reintroduce it in section 2.2.

I am confused by the "Unimodal" vs "Multimodal" and "Older" vs "Younger" labels in Figure 3. To my eye the Ljungan is more prominently bimodal than the Byskealven, yet it plots closer to the Unimodal side of the figure than the Byskealven. Similarly, what portion of the distribution is "Younger" or "Older" when comparing multimodal distributions?

What are the stress values for the MDS plots in figure 3?

---

## Author Comment (AC3)

**Response to Reviewer 1's comments on manuscript egusphere-2022-1200 entitled 'Comparing detrital age spectra, and other geological distributions, using the Wasserstein distance.'**

A. Lipp, P. Vermeesch

January 30, 2023

**Sincerely, I enjoyed reading the manuscript. It is a concise and neatly written technical paper that presents reasoning and implementation of a numerical metric to map age/grains-size distributions in R and Python. The graphical quality of the figures and tables is good. Still, the axes labelling could be improved for readers unfamiliar with multidimensional scaling.**

We thank the reviewer for the encouraging comments.

**Minor point: The title seems to promise more than the manuscript delivers. "Comparing . . . ". The manuscript does not compare something. The presented "comparison" is a performance test of the Wasserstein-2 distance and the Kolmogorov-Smirnov distance. I think the title should reflect better what the manuscript tries to achieve: a presentation of an alternative metric in the realm of multidimensional scaling.**

We believe the reviewer has slightly misunderstood the meaning of our title. Which is meant to describe how the Wasserstein distance can be used to compare distributions as a dissimilarity metric. Not, as the reviewer suggests, that we are comparing the Wasserstein distance itself. We propose to modify the title to make this meaning more clear and prevent confusion. The proposed new title is: *'The Wasserstein distance as a dissimilarity metric for comparing detrital age spectra, and other geological distributions'*.

**The general idea of the manuscript fits within the scope of GChron. However, I am in a little bit of doubt about whether it justifies requesting a peer-review procedure and a peer-review publication. The numerical metric presented here is not new, and the manuscript does not (yet?) show significant scientific progress. The implementation in R appears limited to a few code lines. Perhaps under different circumstances, the implementation in R would have remained a single line in a news files, along with a few lines in the package manual or an entry in a science blog.**

This is a fair comment that could be addressed in two opposite ways:

1. By revising the manuscript as a 'Short Communication' which, according to the *Geochronology* website '*report[s] new developments in or novel aspects of methods, techniques, or tools that are relevant for scientific investigations within the journal scope.*' but are not '*fully self-contained research articles*' could be well suited for this manuscript.

2. By significantly expanding the paper. This option may be unavoidable if we are to address all of Reviewer 2's more substantial comments. These additions will probably extend the paper beyond the length of a 'Short Communication'.

We argue that some form of manuscript, is necessary to communicate the use of this metric to the community. We further argue that the simplicity of what we propose, and its pre-existing implementations are a significant advantage, not a drawback. Both of these mean that its benefits can be immediately and rapidly utilised by the community. The limiting factor is instead a low awareness of the Wasserstein distance in the community. A short communication in a relevant journal is we argue the most appropriate way to overcome this.

**As a non-expert in multidimensional scaling, I feel the manuscript would benefit from more context. The formal description is sufficient and easy to follow, but the likely impact of this manuscript seems low except for having announced a 'new' feature.**

We would be happy to add further context to the Introduction about the importance of dissimilarity measures, in multi-dimensional scaling and elsewise. We strongly contest that the potential impact of this contribution would be 'low'. Since the introduction of multidimensional scaling to detrital geochronology by Vermeesch (2013), the method has been used literally hundreds of times and spawned a small ecosystem of software and derived methods.

How to sensibly evaluate the 'difference' between two observations is of fundamental importance to essentially all statistical methods. Additionally, sensible distance measures are required for developing misfit functions used in inversion schemes (e.g., De Doncker et al. 2020). Whilst MDS is *currently* the most common geochronology technique which uses distance metrics, the potential usage is more significant. The importance of this is reflected in the significant and on-going debate of how best to visualise and compare distributional data (e.g., Sircombe et al. 2004; Saylor et al. 2012; Vermeesch 2012; Satkoski et al. 2013; Vermeesch 2013, 2018; Sundell et al. 2021). We would be happy to incorporate a further discussion along those lines into a revised manuscript.

**In other words: How does this new measure perform for real samples and their (new) interpretation? Section 4 reads interesting, but was a new conclusion reached? Did it lead to better (e.g., more accurate, more precise) results, or did the geoscientific interpretation essentially remains the same? If the latter is the case, perhaps you can present a real case underlining the point you want to make better.**

We agree with the reviewer and will add several new examples. See the beginning of our reply to Reviewer 1 for further details.

**The manuscript comes without a proper discussion. Section 4 is an application example that includes elements of a discussion. However, for a scientific manuscript, I would expect to see more. In particular I would like to see a discussion about the question: Does it likely change the outcome of studies working with this 'new' metric.**

We agree. The revised manuscript will be expanded with an in-depth discussion of the pros and cons of the KS and $W_2$ distances. This will be illustrated with specific examples, as detailed in our response to Reviewer 1. This will give the reader clearer guidance as to when to use which metric.

**The synthetic data outlines the general problem you want to address. I suggest leading with an example based on a case study where the Kolmogorov-Smirnov distance did not perform as expected for the reasons you have mentioned.**

We agree. See examples 1-4 in our response to Reviewer 1.

**Line-by-line comments**

**L111: I've played a bit with the proposed synthetic data and found that it depends to some extent on the standard deviation. A more narrow standard deviation for the same fixed mean values leads to more complex KS-distance patterns. The higher the degree of overlap (higher standard deviation), the more conclusive the KS distance becomes. Perhaps you can add a few lines about it in the text.**

This is an great observation and something we would explore in a revised manuscript.

**L150-L175: I think this paragraph can be improved in order to provide a better experience to readers.**

We are happy to revise this section in line with the constructive feedback provided.

**The R code snippet produces a plot output. However, if I reduce the dataset, it fails. This appears to be a bug in the package 'IsoplotR' because it returns an uncontrolled error:**

```
DZ <- IsoplotR::read.data("scandinavia_short.csv", method = "detritals")
```

We assume that the reviewer has created `scandinavia_short.csv` themself? MDS analysis of two samples is pointless so it is not surprising that this throws an error. It is a bit harsh to call this a bug. A three sample data set does work, however.

**L152: If I look into the R code (file mds.R), I read in line 199 of the code: modified after the wasserstein1d function of the transport package. It is normal to look up open-source code of others, however, if it helped for the own implementation and since the code line in question seems identical, credit should be given in the manuscript to authors of the package 'transport' (Schuhmacher et al. 2022)**

This is a pertinent comment. Proper credit has been given to Schumacher et al. (2022) in the `@author` field of the documentation.

**L164: Please consider adding the example data to the manuscript or the R package**

We would be happy to add the example data as a supplement to the manuscript, in addition to its current location in the GitHub repository.

**L167+ (footnote): The repository pvermees/IsoplotRbeta does not exist, but I guess the branch beta was meant and it should read: remotes::install_github('vermeesch/IsoplotR@beta')**

The reviewer is correct and we would fix this in a revised manuscript.

**Figure 1: How did you modify the data to "aid illustration"? It appears that you have shifted the 'Byskealven' dataset by 1 Ma. This should be mentioned. If so, how does it affect the W1 distance? Your pink area becomes considerably smaller if the non-shifted dataset is used. Perhaps you have a better dataset at hand; one that does not need such manipulation.**

This is correct, the Byskealven sample is simply translated by 1000 Ma, and we will add this to manuscript. This translation adds (exactly) 1000 to $W_1$ between the two samples. Whilst we could seek a 'real' dataset to illustrate this point, we don't see how this would provide any greater insight as by definition, we would be seeking 'real' data that produces the same insights as presently demonstrated. Additionally, introducing too many datasets we believe could be confusing for the reader.

**Figure 2: The upper plot would benefit from y-axis labelling**

We will improve labelling of the MDS figures throughout.

**Figure 3: Something is at odds with the lower figures (j and k), if I try to reproduce them with IsoplotR::mds() and the example data from GitHub. Figure 3j does not look as in the manuscript, because 'Ljusnan' and 'Byskealven' seem to be no different. Figure 3k is somewhat mirrored. Below my code, I first used IsoplotR():mds(), here more manually to show the calculation steps. The mirrored figure is not a big deal because the interpretation should not change, but it should be presented as the users would see it running the code.**

The mirroring is a result of the random state of the MDS algorithm. As this varies from computer to computer, there is little that we can do to prevent this. We will however add text clarifying that reflections are possible but, as the reviewer says, have no impact on any interpretations.

**Additional Comments**

**The following comments refer to something I discovered along the way. It was not considered for my recommendation to the editor.**

We really appreciate the detailed comments and expert suggestions regarding the `IsoplotR` package.

**I have not used 'IsoplotR' before, but it appears to be an interesting and mature package. However, something I noticed missing while searching for the new feature implementation was a NEWS file, as it is custom for R packages. The authors may want to consider adding such a file because it helps users understand package changes without inspecting code changes. This is something I find very useful for scientific software packages.**

Updates were previously listed at `https://isoplotr.es.ucl.ac.uk/home` but have now been added to `NEWS.md` as well. Thank you for the suggestion.

**The R code example returns a default plot (IsoplotR::mds()). Perhaps it is obvious to users familiar with the package, but I found it confusing having numbers on the x and y axis but no axes labels. I guess what is shown is some scaled distance. In Section 4, you have explained it better, but personally, I prefer to understand figures without reading the text.**

The default axis labels have been changed to 'Dim 1' and 'Dim 2'.

**The default plot IsoplotR::mds() returns only the names, but the location in the plot remains a little bit obscure because the default plot setting is pch = NA. I find it awkward because it reduces the value of the plot when what is shown remains too vague to be positioned correctly. In particular, because the text() position in R can be very different from what you would expect depending on the setting in pos.**

The default value of `pch=NA` reduces clutter. However, the default data set in the online GUI uses short labels, which are enclosed in circles. At least 20 times more people use `IsoplotR` via the GUI than do via the command line. Those people who do use the command line tend to know how to change `pch`.

**In the function IsoplotR::plot.MDS(), the standard plot output has the argument asp = 1 hard coded. This is at least unexpected and leads to inconclusive plot behaviour when ylim and xlim are modified**

`asp=1` protects the user from over-interpreting the variations in the second dimension if most of the variability is contained in the first one. One example of this is shown in Figure 3 of the response to Reviewer 2. However, Reviewer 1 is correct about the inconclusive behaviour when both `xlim` and `ylim` are specified. To avoid

this issue, we have added the default argument `asp=1` to the main `mds()` function, where it can be changed by the user.

**References**

De Doncker, F., F. Herman, and M. Fox (2020). "Inversion of provenance data and sediment load into spatially varying erosion rates". *Earth Surface Processes and Landforms* 45.15, pp. 3879–3901.

Satkoski, A. M., B. H. Wilkinson, J. Hietpas, and S. D. Samson (2013). "Likeness among detrital zircon populations—An approach to the comparison of age frequency data in time and space". *GSA Bulletin* 125.11-12, pp. 1783–1799.

Saylor, J., D. Stockli, B. Horton, J. Nie, and A. Mora (2012). "Discriminating rapid exhumation from syndepositional volcanism using detrital zircon double dating: Implications for the tectonic history of the Eastern Cordillera, Colombia". *Bulletin of the Geological Society of America* 124.5-6, pp. 762–779.

Sircombe, K. N. and M. L. Hazelton (2004). "Comparison of detrital zircon age distributions by kernel functional estimation". *Sedimentary Geology.* Quantitative Provenance Analysis of Sediments 171.1, pp. 91–111.

Sundell, K. E. and J. E. Saylor (2021). "Two-Dimensional Quantitative Comparison of Density Distributions in Detrital Geochronology and Geochemistry". *Geochemistry, Geophysics, Geosystems* 22.4, e2020GC009559.

Vermeesch, P. (2012). "On the visualisation of detrital age distributions". *Chemical Geology* 312-313, pp. 190–194.

– (2013). "Multi-sample comparison of detrital age distributions". *Chemical Geology* 341, pp. 140–146.

– (2018). "Dissimilarity measures in detrital geochronology". *Earth-Science Reviews* 178, pp. 310–321.

---

## Author Comment (AC4)

**Response to Reviewer 2's comments on manuscript egusphere-2022-1200 entitled 'Comparing detrital age spectra, and other geological distributions, using the Wasserstein distance.'**

A. Lipp, P. Vermeesch

January 30, 2023

**Summary**

**The authors have not demonstrated that the WS Distance produces geologically meaningful results. At the very least they need to address the concerns raised below before publication. However, the comments below suggest that the WS Distance may not be an appropriate metric to compare geochronological distributions, because of the unique geological implications of minorly distinct age modes (distinct sources) versus multimodal distributions which include some shared age modes (potentially shared sources).**

We thank the reviewer for their critical review, which has prompted us to re-evaluate the pros and cons of the $W_2$ distance compared to the KS-statistic. We agree with the reviewer (below) that '*whether [a dissimilarity measure] is more 'sensible' depends on the application*'. In some cases, this is the $W_2$ distance, but in other cases, the KS statistic may be better. To help the reader make this judgement, we will expand the paper with a thorough discussion of a number of specific scenarios.

The KS-statistic is very good for detecting the *presence* or absence of specific discrete age components. If one does not care about the *size* of the age differences then the KS statistic is generally the best metric. However, even then there are cases where caution must be exercised and the $W_2$ distance is preferred:

1. Studies that combine measurements from several laboratories, which are affected by inter-laboratory *biases* (Košler et al. 2013, Figure 1).

Other common geological scenarios in which the magnitude of the time (/horizontal) axis differences *does* matter include:

2. Studies that combine samples of different depositional age, causing the shape of the source age distributions to change as a function of time (Figure 2).

3. Detrital thermochronology, in which age distributions shift in response to thermal signals (Figure 3).

4. Fitting of detrital age distributions such as mass balance calculations for sediment mixing in a river network (Amidon et al. 2005).

In all of the above cases, the $W_2$ distance is preferred over the KS statistic. Similar considerations apply to other geological distributional data of variables where absolute values (not just theoretical endmembers) are of meaning, notably, grainsize distributions (Weltje et al. 2007).

[Figure]

Figure 1: KDEs (left) and ECDFs (right) of two samples from the inter-laboratory comparison study of Košler et al. (2013), plus a synthetic sample. Dashed lines mark the true ages of the detrital mixture. According to the KS-statistic, the age distribution produced by Lab 4 is more similar to the synthetic distribution than it is to the distribution produced by Lab 1, despite the absence of any shared age components. The $W_2$ distance correctly deems the distribution produced by Lab 4 to be closer to that of Lab 1 than to the synthetic mixture.

[Figure]

Figure 2: MDS configurations using the KS statistic (left) and the $W_2$ distance, following a log transform, (right) of DZ U-Pb data for the Coconino drill core of Gehrels et al. (2020). This core samples, from bottom to top, the following Members of the Chinle Formation: Blue Mesa (black); Lower Sonsela (green); Upper Sonsela (blue); and Petrified Forest (red). Both MDS configurations correctly separate these main subdivisions. However, whereas the KS distance implies that the amount of chronological diversity within the Petrified Forest Member equals that between the Petrified Forest Member and the Blue Mesa Member, the $W_2$ clarifies that the Petrified Forest samples are all quite similar and distinct from the Blue Mesa samples.

[Figure]

Figure 3: KDEs (left) and MDS configurations (middle and right) for a detrital mica $^{40}\text{Ar}/^{39}\text{Ar}$ dataset of Wobus et al. (2003). Samples A–G have been relabelled from the original publication and arranged from north to south across a physiographic transition of the central Himalaya in Nepal. The MDS configuration using the KS statistic (middle) is poorly constrained due to the complete lack of overlap between groups A-D and E-H, respectively. The nearest neighbour lines connect the two groups based on a KS-distance of 1 between samples A and E. The $W_2$ distance (following a log transform) fares much better: it correctly identifies the two groups, which are separated by the physiographic transition. Note that the $W_2$ distance also recovers the correct geographical order of the samples (except for A–D, which are statistically indistinguishable from each other).

**General Comments**

**The WS Distance may be more intuitive than the KS distance in many cases, but whether it is more 'sensible' depends on the application. While the toy dataset clearly shows the advantage of WS over KS for assessing simple dissimilarity between sample ages, in many DZ studies it is the degree to which samples share the same sources that is of interest; the absolute difference in age is not directly relevant. For example, if we assume the sources for the samples in the toy dataset each have distinct ages, A and C are no more similar in terms of their sources than A and D, and the equal KS value of 1 (i.e., complete dissimilarity) for (A, C) and (A, D) is actually more informative than the WS values W(A, C) = 1 and W(A, D) = 10.**

We refer the reviewer/reader to our comments above about scenario 1 in which the $W_2$ is and KS is not the most appropriate dissimilarity measure. We argue that the concern the reviewer raises here (and below) of samples being described as mixtures of very well-defined discrete sources of largely irrelevant absolute ages, is just *one* scenario where measures such as the KS distance would be more appropriate. There are many other scenarios (see above) where it is not correct to discount absolute differences in age and thus where the $W_2$ distance is more appropriate.

**The authors need to explore the behavior of WS Distance for multi-modal data sets. For example, what is the interpretation of high versus low WS Distances when comparing multi-modal data sets? This may seem intuitive, however, I suspect that the results will not be intuitive given how the WS Distance is calculated.**

First, we would like to point out that the KS distance of multimodal datasets does not always yield intuitive results either (see Figure 1). Second, the interpretation of the $W_2$ distance is actually very intuitive. Assuming the two multimodal datasets are aligned, the $W_2$ is simply the cost of 'moving' the grains from one distribution into the shape of another. For example, assuming that the two datasets have the same number of grains, we simply sum the (squared) distance that each grain travels when rearranging them from one distribution along the time axis, to the other. When there are different numbers of grains, some of them are 'split', but conceptually it is the same. If the datasets are misaligned, the $W_2$ is simply an addition of the cost of translating the two datasets to be aligned, and the cost of 'rearranging' the grains. That the $W_2$ is an additive with respect to a) translating distributions and b) changing their shape is intuitive and helpful.

We explore this more fully using a simple synthetic example (Figure 4) where we mix two unimodal distributions at 500 and 800 Ma in proportions between 0 and 1. This generates a suite of mixture distributions with changing levels of bimodality. We then calculate the $W_2$ distance between each of these and the 'central' bimodal distribution. We find that $W_2$ changes nearly linearly with respect to the mixing proportion. We would happily include a similar example to this in the published manuscript.

**The geological context of the samples needs to be provided. Even a reproduction of the Morton et al. (2008) Figure 1 would be helpful in terms of understanding which samples would be predicted to be derived from similar sources.**

Whilst we are happy to provide further context, we note that Reviewer 1 in fact suggests that this table is superfluous. As both reviewers suggest mutually exclusive modifications to this table we suggest that it is adequate in its current form.

**I am fairly surprised that the authors chose samples with n≤60 grains to showcase a new statistical comparison metric. Multiple studies (including one by the co-author) have shown that n≫100 (ideally n>300) are needed for robust statistical comparison. Although it can be argued that samples of that size are unnecessary for simple age distributions such as those presented here, this raises the question of why choose the simplest possible scenario to showcase**

[Figure]

Figure 4: The impact of multi-modal mixing on $W_2$

a new application of a statistical metric. Rather, it seems that a true breakthrough would be demonstrating that the WS Distance can deal with a previously unresolved problem or elegantly deals with an intensely complex data set.

First, whilst we recognise the general importance of considering the number of grains in analyses, it is not relevant in this particular study for the reasons the reviewer already states. Incidentally, we note that, *because it penalises horizontal distances* the Wasserstein distance is actually very robust to potential variations introduced to small sample sizes.

We stress that the Wasserstein distance is indeed useful for comparing 'complex' datasets, as illustrated by the examples provided at the beginning of this response. One interpretation of a dataset's complexity is its dimensionality. As we discuss in the Introduction, the Wasserstein distance in fact extends easily (and elegantly) to distributional data of multiple dimensions. This topic is further discussed in a separate paper that was accepted by another journal (JGR-ES) pending minor revisions (Vermeesch et al. 2023).

**Figure 2 demonstrates problems with both the KS and WS distances. First, the WS distance increases linearly with displacement away from 1000 Ma. However, once the two distributions no longer overlap, they are no longer any more or less similar because the x-axis is age, not distance. Two age distributions that share no age modes are equally dissimilar regardless of the age difference between their modes due to the geological implications of sharing versus not sharing age modes. Hence, the increasing WS Distance with displacement beyond zero overlap is an undesirable trait. Second, both the WS and KS distances indicate that the distributions are most alike (minimum WS and KS distance) when both are centered at 1000 Ma. However, at that point they share no age modes. The green distribution would have an age mode at 1000 Ma, but the black distribution would have age modes at 900 and 1100 Ma. Rather, they should be most alike when the green distribution overlaps with one of the age modes of the black distribution. As an aside, this is the behaviour that cross-correlation of KDEs of these distributions provides (below).**

As discussed above, we agree with the reviewer to an extent. In *some* cases (e.g., when discrete sources are well defined and samples are mixtures of them) non-overlapping samples can all be described as equally dissimilar. As a result, in such a scenario, the KS distance may be preferable. However, as previously discussed, this is just one scenario for which comparing detrital age distributions is used. In many other scenarios, described above, absolute distance along the time/x axis ought be considered for comparing distributions.

**Figure 3 seems problematic for application of the WS distance to detrital geochronology. For example, Ranealven, Lainioalven, and Bysealven all share peaks at 1800 Ma. The KS distance correctly locates these samples closest to each other. The WS distance in contrast locates the Ljusnan closer to the Ranealven than the Lainioalven even though the major age modes between the Ranealven and Lainioalven ( 100 Myr offset) are closer than those of the Ranealvan and Ljusnan ( 200 Myr offset). The WS Distance also locates the Vindelalven and Lainioalven equidistant from the Ranealven. The problem is that distinct source areas may have similar but non-overlapping age modes, and the WS distance is insensitive to these minor differences and therefore unable to discriminate them as distinct sources. A detrital geochronology distribution that has a mode at 1800 Ma and 2800 Ma is more likely to share a source with a sample with a distribution at 1800 Ma (i.e., one overlapping age mode) than one at 1700–1750 Ma (i.e., no overlapping age modes). The authors may be able to address this concern, but it seems to me to be a fatal flaw in this metric.**

The reviewer makes two incorrect statements here. First, '*distinct source areas may have similar but non-overlapping age modes, and the WS distance is insensitive to these minor differences and therefore unable*

*to discriminate them as distinct sources*'. The $W_2$ distance is perfectly sensitive to minor differences in non-overlapping age modes. In fact the $W_2$ between two non-overlapping age modes is *exactly* equal to the age offset between them (as is also shown in Figure 2 and described in Equation 3). We argue that this is in fact, as sensitive (a linear, 1 to 1 relationship) as possible. Thus, not only is the $W_2$ able to discriminate them as distinct sources it *additionally* provides useful geological information on how similar in time those sources are.

Second, the reviewer implies that in the described scenario ('A detrital geochronology distribution that has a mode at 1800 Ma and 2800 Ma is more likely to share a source with a sample with a distribution at 1800 Ma than one at 1700–1750 Ma.'), the Wasserstein distance would be smaller between the 1700 modal sample and the bimodal sample than the 1800 modal sample. This is not correct. We recreate this scenario in Figure 5. The distance matrix between these 3 samples is given in Table 1. As is clear, the distance between samples a & b is smaller than the distance between samples c & b. Consequently, we believe that it is incorrect to suggest that this is a fatal flaw.

However, it is possible to conceive of more extreme scenarios, in which the age difference between the two modes is more extreme (e.g., Cenozoic and Archean). In that case, it is possible that the old age component has an excessive influence on the $W_2$-distance. This problem can be mitigated by log-transforming the data. Incidentally, that is how Figures 2 and 3 were generated. The scale-dependency of the $W_2$ can be rightly considered as a weakness of the method relative to the KS-distance, which is scale invariant. The revised manuscript will be more clear about this limitation.

[Figure]

Figure 5: **Three synthetic distributions.** Two unimodal distributions (gaussian with $\sigma$=100 Ma centred on 1800 Ma (a) and 1700 Ma (b) respectively. One bimodal distribution with peaks at 1800 and 2800 Ma.

|   | a | b | c |
|---|---|---|---|
| a | 0 | 680 | 100 |
| b | 680 | 0 | 757 |
| c | 100 | 757 | 0 |

Table 1: $W_2$ distances between the distributions displayed in Figure 5

**The problems with this metric can be seen when comparing some more complex age distributions. Simple visual inspection indicates that Samples 1, 2, and 7 should be quite similar. This is confirmed by a KS-based MDS (above left), or Kuiper-based MDS (above right). It is also confirmed by a cross-correlation-based MDS (above). However, a WS Distance-based MDS (above) non-intuitively locates samples 1 and 6 quite close, and locates 1, 6, and 7 closer to 5 than to 2.**

It is unclear how best to respond to this comment as what the reviewer states as fact ('Simple visual inspection indicates that Samples 1, 2, and 7 should be quite similar') is actually subjective. Moreover, suggesting that there is one 'correct' sample mapping that a metric can fail to reproduce contradicts with the reviewer's own statement that: 'whether [a distance] is more 'sensible' depends on the application'. Additionally, it is not clear to us whether the desired clustering of 1, 2 & 7 is even reproduced on the MDS maps that the reviewer provides. Whilst in the cross-correlation map, the samples 1,2, & 7 *are* clustered, this is not the case in the KS and Kuiper maps. In the KS map, the distance between samples 1 and 2 is over half the x-range. The same is true for samples 1 and 7 on the Kuiper map.

Notwithstanding the above, we also disagree that the $W_2$ map (Figure 6) is behaving unintuitively. Again we emphasise that there is no one 'correct' mapping of samples. We argue that, however, the mapping produce by $W_2$ is one particularly intuitive mapping that can be deconstructed sequentially in terms of the means and shapes of the distributions. In the MDS shown on Figure 6 we can see that the samples are primarily distributed on a trend from the upper-left to the lower-right (red arrow), giving the ordering: 2, 7, 1, 6, 5, 4 & 3. We now visualise the KDEs of these samples sequentially in this order on the left-hand (red) column on Figure 6. We observe that this ordering coincides with the ordering from youngest to oldest of the average age of the grains (highlighted with vertical grey dashed lines). We next investigate the ordering of samples on the perpendicular direction from the top-right to the lower-left (blue arrow). Visualising samples in this order (right-hand column, note that these samples have been mean-shifted) we can see that this corresponds to a change in shape from bimodal to unimodal distributions. As a result, we see that, counter to the Reviewer's suggestion, $W_2$ MDS maps often contain latent directions which are easily interpreted. A similar analysis using the KS distance does not produce readily interpretable latent directions (Figure 7).

**A second example highlights the disproportionate impact of misalignment of detrital age modes on the WS Distance. P1 and P3 below share age modes at 1800 Ma. P3 has an additional mode at 500 Ma. P2 shares no age modes with either P1 or P3 and rather has an age mode at 1700 Ma. This is reflected in their KS MDS plot (above left) and an MDS plot based on cross-correlation (above right, note difference in x- and y-axis scale). However, the WS Distance MDS (above) does not reflect the true relationship between age modes, but rather reflects the anomalous area between P1 and P3 (i.e., the horizontal distance between 500 Ma and 1800 Ma in the ECDF plot).**

We disagree that misalignment of detrital modes has a **dis**proportionate impact on $W_2$. As discussed above, the misalignment of modes is in fact (literally) proportional to the $W_2$ distance. In this particular example we again emphasise that there is no 'correct' mapping for a set of samples. As discussed above, this particular analysis is one where the KS distance may be appropriate, as the absolute ages are not of major importance. However, we reiterate that the mapping produced by $W_2$ still follows a consistent logic, which considers the absolute ages of the samples. As P3 is the only sample with any grains derived from 'young' sources, it is identified as an outlier. Such a grouping could be intuitive for many (but not all) uses such as identifying approximate ages of deposition/formation. Secondly, we note again that this mapping still distinguishes samples P1 and P2, with a $W_2$ between them of 100, corresponding to the offset of their peaks. Such an interpretation may not be applicable for *every* study, but it is up to the user to determine which metric is most appropriate.

[Figure]

Figure 6: Top figure shows MDS map using $W_2$ distance as metric of samples provided by reviewer. Arrows indicate the 'order' of samples which are visualised in KDEs in the columns below. Left column shows order of samples from top-left to bottom-right. Right column (mean shifted) shows order from top-right to bottom-left.

[Figure]

Figure 7: Same as Figure 6 but using KS distance

**Detailed comments**

**The authors assert that cross-correlation is an ad hoc method, yet the Pearson coefficient which is the basis for the cross-correlation coefficient is widely used in seismic analysis, waveform analysis, and image analysis. As such, it is unclear what is meant by "ad hoc" in this context. Although Vermeesch (2018) correctly points out limitations of cross-correlation as applied to probability density plots of extremely precise data, these caveats do not apply to its application to kernel density estimates or kernel functional estimates. Similarly, the charge that Likeness is ad hoc is unfounded. Likeness is an adaptation of the L1 norm applied, usually, to a 1D geochronology distribution. (However, see Sundell et al. (2021) for application of the L1 norm to 2D distributions.) Finally, like the cross-correlation coefficient and Likeness, the Sircombe-Hazelton distance (L2 norm, Sircombe and Hazelton, 2004; Vermeesch, 2018) also requires discretization of continuous functions for its calculation.**

The application of cross-correlation to age distributions looks superficially similar to the applications in seismology and image analysis, but is fundamentally different. In seismology, acoustic amplitude is recorded at regular time intervals. In digital image analysis, pixels in a CCD are regularly spaced in a grid. However, the U-Pb ages that constitute a detrital spectrum are not regularly spaced. Unlike the Kolmogorov-Smirnov and Wasserstein distances, which readily accept these irregular data, the cross-correlation and likeness coefficients require that the data are 'shoehorned' into an evenly spaced set of intervals.

The proponents of these methods argue that this can be done using Probability Density Plots (PDPs) or Kernel Density Estimates (KDEs). The shortcomings of PDPs were pointed out by Vermeesch (2012) and are acknowledged by the reviewer. KDEs are also problematic, as they require the selection of a bandwidth. Different bandwidths result in different cross-correlation coefficients and likeness factors. We do not see any justification for this arbitrary intermediate step, which is not required by the KS and $W_2$ distances.

The cross-correlation and likeness metrics are 'ad hoc' methods for the same reason why the PDP is an ad hoc method. PDPs superficially look like KDEs (Brandon, 1996) but are fundamentally different. Similarly, the cross-correlation coefficient superficially looks like its equivalent in waveform analysis and image recognition, but it is fundamentally different. And the Likeness factor looks a bit like Sircombe et al. (2004)'s L2 norm, but serves a completely different purpose. The S-H metric is used in a niche application, whereby high precision and low precision datasets are combined. In this specific case, some smoothing is justified (and even necessary). In most situation, this is not the case and smoothing should be avoided.

**I guess the take away from this is that it may be useful to compare the performance of cross-correlation and Likeness in addition to the KS Distance to newly applied metrics like the WS Distance.**

Vermeesch (2018) argued that cross-correlation and likeness should be abandoned, so we will not discuss them further in our paper.

**It is ironic that after railing against use of the cross-correlation coefficient, the authors reintroduce it in section 2.2.**

$\rho^{\mu\nu}$ is *not* a cross-correlation coefficient. It compares quantiles, not density estimates.

**I am confused by the "Unimodal" vs "Multimodal" and "Older" vs "Younger" labels in Figure 3. To my eye the Ljungan is more prominently bimodal than the Byskealven, yet it plots closer to the Unimodal side of the figure than the Byskealven. Similarly, what portion of the distribution is "Younger" or "Older" when comparing multimodal distributions?**

'Younger' and 'older' in this Figure refers to the mean age of the samples (c.f., Figure 6). In a revised manuscript we will clarify the description of this figure, in a style similar to those presented above (e.g., Figure 6)

**What are the stress values for the MDS plots in figure 3?**

We are happy to add the stress values to the MDS plots in a revised manuscript. In nearly all cases that we have seen, the $W_2$ distance leads to lower stress values than the KS-statistic. However this does not, in itself, mean that $W_2$ is better than KS. It just means that $W_2$-distances are more easily captured by two-dimensional MDS configurations than KS-distances are. The reason for this is apparent in the examples of Figures 2 and 3. The $W_2$-based MDS configurations are almost one-dimensional patterns, whereas the KS-based configurations fill the 2D-space.

**References**

Amidon, W. H., D. W. Burbank, and G. E. Gehrels (2005). "Construction of detrital mineral populations: insights from mixing of U-Pb zircon ages in Himalayan rivers". *Basin Research* 17, pp. 463–485.

Gehrels, G., D. Giesler, P. Olsen, D. Kent, A. Marsh, W. Parker, C. Rasmussen, R. Mundil, R. Irmis, J. Geissman, et al. (2020). "LA-ICPMS U–Pb geochronology of detrital zircon grains from the Coconino, Moenkopi, and Chinle Formations in the Petrified Forest National Park (Arizona)". *Geochronology*.

Košler, J., J. Sláma, E. Belousova, F. Corfu, G. E. Gehrels, A. Gerdes, M. S. A. Horstwood, K. N. Sircombe, P. J. Sylvester, M. Tiepolo, M. J. Whitehouse, and J. D. Woodhead (2013). "U-Pb Detrital Zircon Analysis – Results of an Inter-laboratory Comparison". *Geostandards and Geoanalytical Research* 37.3, pp. 243–259.

Sircombe, K. N. and M. L. Hazelton (2004). "Comparison of detrital zircon age distributions by kernel functional estimation". *Sedimentary Geology* 171, pp. 91–111.

Vermeesch, P., A. Lipp, D. Hatzenbühler, L. Caracciolo, and D. Chew (2023). "Multidimensional scaling of varietal data in sedimentary provenance analysis". *Journal of Geophysical Research – Earth Surface*.

Vermeesch, P. (2012). "On the visualisation of detrital age distributions". *Chemical Geology* 312-313, pp. 190–194.

– (2018). "Dissimilarity measures in detrital geochronology". *Earth-Science Reviews* 178, pp. 310–321.

Weltje, G. J. and M. A. Prins (2007). "Genetically meaningful decomposition of grain-size distributions". *Sedimentary Geology*. From Particle Size to Sediment Dynamics 202.3, pp. 409–424.

Wobus, C. W., K. V. Hodges, and K. X. Whipple (2003). "Has focused denudation sustained active thrusting at the Himalayan topographic front?" *Geology* 31, pp. 861–864.

---

## Author Response (AR1)

**Author's response**

A. Lipp, P. Vermeesch

March 13, 2023

**Associate Editor**

**Regarding comments by referee 1, the change of title seems justified. I also see that the manuscript aligns well with the "Short Communication" scope of the journal. The suggested expansion of contextual information would be helpful as short "onboarding" in the introduction or, maybe, the discussion. This needs to be evaluated after it has been implemented. The suggested actions to the line by line comments appear appropriate to me and I look forward to seeing the updated version of the manuscript in due course.**

We have resubmitted the manuscript as a 'Short Communication' as proposed. Additionally, we have revised the Introduction to better provide context on MDS, as well as the importance of dissimilarity metrics more generally. Additionally we have modified the manuscript according to the reviewer's line-by-line comments.

**Regarding comments by referee 2, I see that a large portion of the discourse arises from different perspectives on the same topic, which usually means that the clarity of the text needs to be improved in order to resolve misunderstandings from the beginning. I suggest to add clear statements about boundary conditions, assumptions and propositions early on (e.g., proposed subjectivity of visual inspection, non-unique sources for similar distribution patterns, etc.), to set the right expectation space for future readers. I would welcome a slightly more profound reevaluation of pros and cons as suggested by referee(s) and authors. In addition, it may help to clearly discuss the extreme endmember states of data and how the metrics behave for those, but also to discuss how the space in between such endmember states affects the KS and W2 results. I agree that the table 1 is of limited value in its current form. Perhaps replacing it by a 1-3 sentence verbal (and referenced) description of the geological context would be a useful way to provide the required background. I would encourage the authors to briefly add the the information on minimum sample size explicitly, to clarify that this has not been overlooked.**

In response to the constructive comments from reviewer 2 we have made substantial revisions to the manuscript. First, we now emphasise that there is no 'correct' dissimilarity metric for all scenarios. We have added the following to the Introduction, and similar text is added in the Discussion section:

> For all uses, the choice of which dissimilarity metric to use is vital as different metrics result in different numerical results and thus different geological interpretations. In general, the most appropriate metric will depend on the data being analysed and the scientific question under investigation.

Second, more substantially, we have added a new Discussion section (Section 3) which carefully weighs up and advantages and disadvantages of $W_2$ in various geochronological scenarios. In this section we present four different case studies, (three real, one synthetic) discussing whether the $W_2$ is preferable to the KS in each. We identify a 'rule-of-thumb' that if absolute age differences are deemed important information for the problem, $W_2$ is to be preferred. Elsewise, the KS distance may be preferable and the KS distance can be

unintuitive. This final scenario applies in cases of mixing of discrete endmembers, as identified by Reviewer 2. This new section is copied below.

Finally, due to the addition of these new examples, we decided that the example from Morton et al. (2008) was redundant, and has been removed from the manuscript.

The new Discussion section:

[revised manuscript text omitted]

**Reviewer 1**

**Minor point: The title seems to promise more than the manuscript delivers. "Comparing . . . ". The manuscript does not compare something. The presented "comparison" is a performance test of the Wasserstein-2 distance and the Kolmogorov-Smirnov distance. I think the title should reflect better what the manuscript tries to achieve: a presentation of an alternative metric in the realm of multidimensional scaling.**

To make the title more clear we have revised the manuscript title to now read as: *'Short Communication: The Wasserstein distance as a dissimilarity metric for comparing detrital age spectra, and other geological distributions'*

**The general idea of the manuscript fits within the scope of GChron. However, I am in a little bit of doubt about whether it justifies requesting a peer-review procedure and a peer-review publication. The numerical metric presented here is not new, and the manuscript does not (yet?) show significant scientific progress. The implementation in R appears limited to a few code lines. Perhaps under different circumstances, the implementation in R would have remained a single line in a news files, along with a few lines in the package manual or an entry in a science blog.**

As proposed, we are re-submitting the manuscript as a 'Short Communication'.

**As a non-expert in multidimensional scaling, I feel the manuscript would benefit from more context. The formal description is sufficient and easy to follow, but the likely impact of this manuscript seems low except for having announced a 'new' feature.**

We have now revised the introduction to improve the background coverage of MDS and also to better emphasise the importance of dissimilarity measures. The new introductory paragraph reads as follows:

> A distributional dataset is one where the information does not lie in individual observations, but in the *distribution* of many observations associated with one sample. Such data are common in the geological sciences, for example, detrital mineral ages or grain size distributions. Zircon U-Pb ages, in igneous and detrital samples, are one particularly widely used class of distributional data, which are used *inter alia* to constrain sediment provenance, global magmatic processes, and the evolution of plate tectonics (e.g., Condie et al. 2009; Cawood et al. 2012; Reimink et al. 2021). Grainsize distributions are another common form of geological distributional data. Analytical advances mean that increasingly large amounts of distributional data are being generated in the Earth sciences meaning that qualitative comparison of samples is becoming infeasible, and objective dissimilarity metrics between samples must be used. Some measure of dissimilarity (or more specifically, distance) is also required for many widely used statistical methods such as clustering, ANOVA, and dimension reduction. Dissimilarity metrics in geochronology at present are most commonly used for dimension reducing techniques such as multi-dimensional scaling (MDS) or principal component analysis (PCA). Such methods have become popular for analysing large numbers of detrital age spectra simultaneuously (Vermeesch 2013; Sharman et al. 2018; Vermeesch 2018a). Fitting models (e.g., sediment source partitioning) using distributional data also requires a definition of dissimilarity for comparing observed and predicted distributions (e.g., Amidon et al. 2005; De Doncker et al. 2020).

**In other words: How does this new measure perform for real samples and their (new) interpretation? Section 4 reads interesting, but was a new conclusion reached? Did it lead to better (e.g., more accurate, more precise) results, or did the geoscientific interpretation essentially**

**remains the same? If the latter is the case, perhaps you can present a real case underlining the point you want to make better.**

In the new Discussion section (Section 3 in the revised manuscript) we prevent three real and one synthetic example where the behaviour of KS and $W_2$ are evaluated in detail. We show that in the examples from DeGraaff-Surpless et al. (2002), Wobus et al. (2003), and Košler et al. (2013), the KS distance provides unsatisfying solutions (and inaccurate in the case of Košler et al. 2013) whereas the $W_2$ distance better captures geological intuition.

**The manuscript comes without a proper discussion. Section 4 is an application example that includes elements of a discussion. However, for a scientific manuscript, I would expect to see more. In particular I would like to see a discussion about the question: Does it likely change the outcome of studies working with this 'new' metric.**

As mentioned in the response to the previous comment we have now added a Discussion section (Section 3) which discusses four different scenarios where the use of the $W_2$ distance may (or may not) improve the outcome of different geochronological studies.

**The synthetic data outlines the general problem you want to address. I suggest leading with an example based on a case study where the Kolmogorov-Smirnov distance did not perform as expected for the reasons you have mentioned.**

We believe this has been resolved by the inclusion of three examples in the Discussion section in which the KS distance performs in an unexpected manner, in particular using the data of Košler et al. (2013), where the KS distance fails to identify two theoretically identical synthetic samples as being similar.

**Line-by-line comments**

**L111: I've played a bit with the proposed synthetic data and found that it depends to some extent on the standard deviation. A more narrow standard deviation for the same fixed mean values leads to more complex KS-distance patterns. The higher the degree of overlap (higher standard deviation), the more conclusive the KS distance becomes. Perhaps you can add a few lines about it in the text.**

Whilst we agree that this is a useful observation we could not identify an immediate geological parallel for this specific behaviour, and as such felt it may be beyond the scope of the manuscript.

**L150-L175: I think this paragraph can be improved in order to provide a better experience to readers.**

This section has been revised to now read as follows:

> Additionally, the $W_2$-distance has been added to the `IsoplotR` package in `R`, which calculates dissimilarity matrices and MDS maps (Vermeesch 2018b). This software can be accessed using an (online) graphical user interface, at `isoplotr.es.ucl.ac.uk`. Alternatively, the function can also be accessed from the `R` command line. The following snippet uses $W_2$ to calculate an MDS map for the dataset from Wobus et al. (2003) discussed in the manuscript (Figure 5). The data required is also available at the above repository. Note that the MDS map produced may show slight differences to those in the manuscript due to dependence of metric MDS on a random state variable.

```
**load the package:**
library(IsoplotR)
DZ <- read.data("wobus.csv",method="detritals")
**example 1. calculate the W2 distance matrix for the dataset:**
d <- diss(DZ,method="W2")
**example 2. apply MDS to the dataset:**
mds(DZ,method="W2")
```

**L164: Please consider adding the example data to the manuscript or the R package**

The example data has been added to the data repository.

**L167+ (footnote): The repository pvermees/IsoplotRbeta does not exist, but I guess the branch beta was meant and it should read: remotes::install_github('vermeesch/IsoplotR@beta')**

The link has been corrected to: `isoplotr.es.ucl.ac.uk`.

**Figure 1: How did you modify the data to "aid illustration"? It appears that you have shifted the 'Byskealven' dataset by 1 Ma. This should be mentioned. If so, how does it affect the W1 distance? Your pink area becomes considerably smaller if the non-shifted dataset is used. Perhaps you have a better dataset at hand; one that does not need such manipulation.**

As discussed in the previous response to Authors we not believe that modifying this Figure provides greater geological insight.

**Figure 2: The upper plot would benefit from y-axis labelling**

We have made the axis labelling of MDS plots consistent throughout the manuscript.

**Figure 3: Something is at odds with the lower figures (j and k), if I try to reproduce them with IsoplotR::mds() and the example data from GitHub. Figure 3j does not look as in the manuscript, because 'Ljusnan' and 'Byskealven' seem to be no different. Figure 3k is somewhat mirrored. Below my code, I first used IsoplotR():mds(), here more manually to show the calculation steps. The mirrored figure is not a big deal because the interpretation should not change, but it should be presented as the users would see it running the code.**

We have added the following text to the 'Implementation' section (Line 199) to emphasise that results of MDS calculations may vary: '*Note that the MDS map produced may show slight differences to those in the manuscript due to dependence of metric MDS on a random state variable.*'

**Reviewer 2**

**The authors have not demonstrated that the WS Distance produces geologically meaningful results. At the very least they need to address the concerns raised below before publication. However, the comments below suggest that the WS Distance may not be an appropriate metric to compare geochronological distributions, because of the unique geological implications of minorly distinct age modes (distinct sources) versus multimodal distributions which include some shared age modes (potentially shared sources).**

We agree with the reviewer that in this particular scenario (mixing of discrete and known sources) $W_2$ may not be preferable, and now discuss this, with an example, in the new Section 3.1 of the manuscript. It is presented along with three other scenarios where $W_2$ is found to be preferable to the KS distance. This new section is copied above.

**The WS Distance may be more intuitive than the KS distance in many cases, but whether it is more 'sensible' depends on the application. While the toy dataset clearly shows the advantage of WS over KS for assessing simple dissimilarity between sample ages, in many DZ studies it is the degree to which samples share the same sources that is of interest; the absolute difference in age is not directly relevant. For example, if we assume the sources for the samples in the toy dataset each have distinct ages, A and C are no more similar in terms of their sources than A and D, and the equal KS value of 1 (i.e., complete dissimilarity) for (A, C) and (A, D) is actually more informative than the WS values W(A, C) = 1 and W(A, D) = 10.**

We agree with the reviewer that $W_2$ may not be relevant for all scenarios. As a result we have now added the new Discussion section to consider cases where $W_2$ may or may not be preferred. We explicitly describe how $W_2$ is preferable when absolute age differences are useful information, and the KS distance, when such information is not relevant. For example on Line 143 we now state:

> In general, the Wasserstein distance is most appropriate when absolute differences along the time axis (or more generally, the x-axis) provide useful information to solving the geologic problem. The KS distance however is more appropriate when the size of the time differences between peaks is not relevant.

**The authors need to explore the behavior of WS Distance for multi-modal data sets. For example, what is the interpretation of high versus low WS Distances when comparing multimodal data sets? This may seem intuitive, however, I suspect that the results will not be intuitive given how the WS Distance is calculated.**

In Section 3.2 ('Temporally varying source age distributions') we discuss how $W_2$ is able to accurately distinguish multi-modal samples from uni-modal ones. We note that in this particular example, the KS distance is less effective at identifying the change in distribution shape with stratigraphic height. This section is copied above.

**The geological context of the samples needs to be provided. Even a reproduction of the Morton et al. (2008) Figure 1 would be helpful in terms of understanding which samples would be predicted to be derived from similar sources.**

Given that we have now added four more examples (three with real data), we decided that this original example from Morton et al. (2008) was no longer needed and have removed it from the revised manuscript.

**I am fairly surprised that the authors chose samples with n≤60 grains to showcase a new statistical comparison metric. Multiple studies (including one by the co-author) have shown**

that n≫100 (ideally n>300) are needed for robust statistical comparison. **Although it can be argued that samples of that size are unnecessary for simple age distributions such as those presented here, this raises the question of why choose the simplest possible scenario to showcase a new application of a statistical metric. Rather, it seems that a true breakthrough would be demonstrating that the WS Distance can deal with a previously unresolved problem or elegantly deals with an intensely complex data set.**

We have removed the example from Morton et al. (2008) from the manuscript. As such, whilst we agree with the reviewer about the importance of sufficient numbers of grains, this comment is no longer relevant.

In the new Discussion section (Section 3), we provide three examples where the $W_2$ distance is better able to extract geological information from distributional data than the KS distance. Note however, that we also include an example (Section 3.1) that discusses where the $W_2$ sample may *not* provide greater insight over the KS distance.

**Figure 2 demonstrates problems with both the KS and WS distances. First, the WS distance increases linearly with displacement away from 1000 Ma. However, once the two distributions no longer overlap, they are no longer any more or less similar because the x-axis is age, not distance. Two age distributions that share no age modes are equally dissimilar regardless of the age difference between their modes due to the geological implications of sharing versus not sharing age modes. Hence, the increasing WS Distance with displacement beyond zero overlap is an undesirable trait. Second, both the WS and KS distances indicate that the distributions are most alike (minimum WS and KS distance) when both are centered at 1000 Ma. However, at that point they share no age modes. The green distribution would have an age mode at 1000 Ma, but the black distribution would have age modes at 900 and 1100 Ma. Rather, they should be most alike when the green distribution overlaps with one of the age modes of the black distribution. As an aside, this is the behaviour that cross-correlation of KDEs of these distributions provides (below).**

We agree with the reviewer that in some scenarios (including the one indicated by the the reviewer) the $W_2$ may not be appropriate. In the new Discussion of the manuscript (see Section 3.1) we now consider a case where absolute distance along the time-axis may not be useful information, and as a result $W_2$ behaves unintuitively.

**Figure 3 seems problematic for application of the WS distance to detrital geochronology. For example, Ranealven, Lainioalven, and Bysealven all share peaks at 1800 Ma. The KS distance correctly locates these samples closest to each other. The WS distance in contrast locates the Ljusnan closer to the Ranealven than the Lainioalven even though the major age modes between the Ranealven and Lainioalven ( 100 Myr offset) are closer than those of the Ranealvan and Ljusnan ( 200 Myr offset). The WS Distance also locates the Vindelalven and Lainioalven equidistant from the Ranealven. The problem is that distinct source areas may have similar but non-overlapping age modes, and the WS distance is insensitive to these minor differences and therefore unable to discriminate them as distinct sources. A detrital geochronology distribution that has a mode at 1800 Ma and 2800 Ma is more likely to share a source with a sample with a distribution at 1800 Ma (i.e., one overlapping age mode) than one at 1700–1750 Ma (i.e., no overlapping age modes). The authors may be able to address this concern, but it seems to me to be a fatal flaw in this metric.**

As discussed above, this example has been removed from the manuscript.

**The problems with this metric can be seen when comparing some more complex age distri-**

butions. Simple visual inspection indicates that Samples 1, 2, and 7 should be quite similar. This is confirmed by a KS-based MDS (above left), or Kuiper-based MDS (above right). It is also confirmed by a cross-correlation-based MDS (above). However, a WS Distance-based MDS (above) non-intuitively locates samples 1 and 6 quite close, and locates 1, 6, and 7 closer to 5 than to 2.

As discussed in the previous Author's comment we do not believe that in the example provided the $W_2$ behaved unintuitively. We now mention twice (Lines 34 & 129) that there is no one correct dissimilarity metric and that different metrics will sometimes produce different results. We believe the new Discussion section will equip readers to decide whether the $W_2$ is appropriate for any particular scenario.

A second example highlights the disproportionate impact of misalignment of detrital age modes on the WS Distance. P1 and P3 below share age modes at 1800 Ma. P3 has an additional mode at 500 Ma. P2 shares no age modes with either P1 or P3 and rather has an age mode at 1700 Ma. This is reflected in their KS MDS plot (above left) and an MDS plot based on cross-correlation (above right, note difference in x- and y-axis scale). However, the WS Distance MDS (above) does not reflect the true relationship between age modes, but rather reflects the anomalous area between P1 and P3 (i.e., the horizontal distance between 500 Ma and 1800 Ma in the ECDF plot).

We agree with the reviewer that the $W_2$ distance can be unintuitive in scenarios of mixing between well defined age peaks. This limitation is discussed in the new Discussion Section (specifically, Section 3.1) of the manuscript, copied above.

The authors assert that cross-correlation is an ad hoc method, yet the Pearson coefficient which is the basis for the cross-correlation coefficient is widely used in seismic analysis, waveform analysis, and image analysis. As such, it is unclear what is meant by "ad hoc" in this context. Although Vermeesch (2018) correctly points out limitations of cross-correlation as applied to probability density plots of extremely precise data, these caveats do not apply to its application to kernel density estimates or kernel functional estimates. Similarly, the charge that Likeness is ad hoc is unfounded. Likeness is an adaptation of the L1 norm applied, usually, to a 1D geochronology distribution. (However, see Sundell et al. (2021) for application of the L1 norm to 2D distributions.) Finally, like the cross-correlation coefficient and Likeness, the Sircombe-Hazelton distance (L2 norm, Sircombe and Hazelton, 2004; Vermeesch, 2018) also requires discretization of continuous functions for its calculation. I guess the take away from this is that it may be useful to compare the performance of cross-correlation and Likeness in addition to the KS Distance to newly applied metrics like the WS Distance.

It is ironic that after railing against use of the cross-correlation coefficient, the authors reintroduce it in section 2.2.

Detailed responses to these comments are provided in the previous author comments.

I am confused by the "Unimodal" vs "Multimodal" and "Older" vs "Younger" labels in Figure 3. To my eye the Ljungan is more prominently bimodal than the Byskealven, yet it plots closer to the Unimodal side of the figure than the Byskealven. Similarly, what portion of the distribution is "Younger" or "Older" when comparing multimodal distributions?

This example has been removed the manuscript.

---

## Referee Report (RR2)

**Re-review - Geochronology**

invitation received: 2023-03-14 | today: 2023-03-27

**1 General comment**

After having carefully read the responses, all reviewer comments, and authors' responses, as well as the revised version of the manuscript and I am happy to write that I can support a publication of the manuscript.

The authors have addressed all major concerns and clarified their manuscript where required. The responses are acceptable throughout, and the advantage of using the $W_2$ distance comes across more clearly. Below I have listed a few minor comments, nothing that would block the green light from my side, however.

**2 Response to author comments**

> We strongly contest that the potential impact of this contribution would be 'low'. Since the introduction of multidimensional scaling to detrital geochronology by Vermeesch (2013), the method has been used literally hundreds of times and spawned a small ecosystem of software and derived methods.

I feel that this is the only author's comment where a response is needed because my initial comment was oddly received. I did not contest or intend to diminish previous work. The presented work will add a feature to an existing toolbox, and the revised manuscript, with its discussion makes a justified contribution to the field. However, to me, it was and still is, more of a brief review of an existing metric one can use additionally under particular circumstances. There is nothing wrong with it, but we should also be realistic about the potential impact the presented manuscript will have.

**3 Minor comments**

- L8: Perhaps, "... as an additional and alternative metric ..." to emphasise add-on character of the metric.

- L190 and later: please decide whether to use mono space letters of Python and R or not. I suggest using the mono space letters for source code only. The canonical writing capitalises the first letter of "Python", although the Python logo uses a small "p".

- L188: Please set URLs as (valid) links throughout (also, e.g., L194-195)

- Figures 1-6: Please unify the axis labels (e.g., "Age, Ma" vs "age [Ma]"). Ma should not read "ma"

- Figure 5:
  - Figure caption: add white space between "a)" and KDE.
  - Furthermore, it would be beneficial if the R code reflects the code used to create the figure (not all of it but the $W_2$ part. Currently it looks a lot different from what you have in the manuscript (even if I play with the other graphical parameters); nothing related to the random state parameters. The online GUI shows the same result.
  - If you feel that the log conversion is useful (I think it is) you should add it to `IsoplotR::mds()` or `IsoplotR::read.data()`.
  - Please add more white space to the code snippets (as shown below), it is just a more reader friendly code formating

```r
library(IsoplotR)
**load the package: 200 library(IsoplotR)**
DZ <- read.data("wobus.csv", method = "detritals")
for(i in 1:length(DZ)) DZ[[i]] <- log(DZ[[i]])

**example 1. calculate the W2 distance matrix for the dataset:**
d <- diss(DZ, method = "W2")
```

```
**[1] "W2"**
```

```r
**example 2. apply MDS to the dataset:**
mds(DZ, method = "W2", pch = 20)
```

```
**[1] "W2"**
**initial  value 0.000000**
**final  value 0.000000**
**converged**
```

- L266-268: Please check the reference, it does not look right. The correct citation entry of R packages should be generated using `utils::citation("<R package>")`

- L283: I believe that the series number is missing.

---

## Author Response (AR2)

**Response to second round of reviewer comments**

A. Lipp, P. Vermeesch

April 3, 2023

**Associate Editor**

**I have read the updated manuscript and the referee's letters. I believe that accounting for the one remaining issue of referee 1 and fixing the few remaining issues will move the manuscript to a state where it will become a valuable contribution to the journal's portfolio.**

We're pleased to hear this positive feedback. We have responded to all of the reviewers' remaining comments below and, where appropriate, modified the manuscript accordingly.

**Reviewer 1**

**I feel that this is the only author's comment where a response is needed because my initial comment was oddly received. I did not contest or intend to diminish previous work. The presented work will add a feature to an existing toolbox, and the revised manuscript, with its discussion makes a justified contribution to the field. However, to me, it was and still is, more of a brief review of an existing metric one can use additionally under particular circumstances. There is nothing wrong with it, but we should also be realistic about the potential impact the presented manuscript will have.**

We apologise for any miscommunication, and did not intend to be provide an odd response. We simply feel strongly that solid quantitative foundations are important to the field. Whilst the reviewer we are sure did not intend to diminish previous work, I'm sure they can understand our motivation for making sure that it is not neglected. We hope that the recasting the manuscript as a 'Short Communication' best suits the nature of this study, and does not seek to 'overstate' potential impacts.

**Minor Comments**

**L8: Perhaps, '...as an additional and alternative metric...' to emphasise add-on character of the metric.**

We have added this to the abstract as suggested.

**L190 and later: please decide whether to use mono space letters of Python and R or not. I suggest using the mono space letters for source code only. The canonical writing capitalises the first letter of "Python", although the Python logo uses a small "p".**

Thank you for raising this. We have removed mono spacing for referring to programming languages and consistently use a capital 'P' for Python.

**L188: Please set URLs as (valid) links throughout (also, e.g., L194-195)**

This has been amended as suggested.

**Figures 1-6: Please unify the axis labels (e.g., 'Age, Ma' vs 'age [Ma]'). Ma should not read "ma"**

Thank you for pointing this out, it has been revised as proposed.

**Figure 5**

- **Figure caption: add white space between "a)" and KDE.**

- This has been added.

- **Furthermore, it would be beneficial if the R code reflects the code used to create the figure (not all of it but the W2 part. Currently it looks a lot different from what you have in the manuscript (even if I play with the other graphical parameters); nothing related to the random state parameters. The online GUI shows the same result.**

- Unfortunately its unclear to us how this could be improved as, barring a rotation, the structure presented by the reviewer are nearly identical to those we present in Figure 5. We do not see how the results presented here result in different interpretations. We have clarified however that arbitrary rotations and translations (which do not affect the interpretations) are to be expected.

- **If you feel that the log conversion is useful (I think it is) you should add it to IsoplotR::mds() or IsoplotR::read.data().**

- It will be considered as an option for the MDS functions in future versions, thank you for raising this.

**Please add more white space to the code snippets (as shown below), it is just a more reader friendly code formatting**

We have added in white space to the code snippets as suggested.

**L266-268: Please check the reference, it does not look right. The correct citation entry of R packages should be generated using utils:citation(...)**

Thank you for raising this, we have corrected this.

**L283: I believe that the series number is missing.**

We have corrected the reference here.

**Reviewer 2**

**This is now a second review of the manuscript submitted by Lipp and Vermeesch. The authors have addressed many issues raised by the other reviewer and myself. They have significantly expanded the scope of the discussion and included examples of where the Wasserstein distance both works well and fails. This version is significantly improved over the previous one. My remaining reservations with the manuscript address the underlying method of calculating the Wasserstein distance rather than its application as presented in the manuscript. With the exception of item 1 below, I think the manuscript could be published with minor revisions. I**

would like to see the authors address item 1 however.

We are pleased to here the reviewer believes the manuscript has been significantly improved on the basis of their constructive feedback. We recognise the concern motivating their first item, and agree that Figure 2 could be potentially confusing if coming from a perspective of aligning peaks. Consequently we have addressed their concern by adding further clarifying text to the manuscript, as they suggest. This response, and others, are detailed below.

**1. I maintain that the example in Figure 2 is problematic for both the KS and Wasserstein distances. As stated in the initial review, the translated distribution is most similar to the fixed distribution when their peaks align at 900 Ma or 1100 Ma. When the translated distribution is centered at 1000 Ma, it shares no ages with the fixed distribution (albeit the tails of the distributions overlap). Nevertheless, both the KS and Wasserstein distances indicate that at this point the distributions are most similar. The authors have not addressed this problem either in their response or in the text. Surely at the very least, some explanation for why the behaviour observed in the KS and Wasserstein distances is desirable or intuitive is in order to justify using this example (or these metrics).**

We feel that *'the translated distribution is most similar to the fixed distribution when their peaks align at 900 Ma or 1100 Ma'* is not an objective fact, but a matter of subjectivity. As we now discuss in the main manuscript, there is no single distance function that could be applied in all scenarios. This logic extends to this figure too. As a result, we do not believe that this is really a 'problem' that needs to be addressed as suggested. Additionally, we believe the reviewer is neglecting the fact that both metrics do not suggest that, in absolute terms, the central value is a perfect fit. Both $W_2$ and KS are non-zero at the central point, indicating correctly that the unimodal and bimodal distributions are different. As such we don't feel that they 'fail' in this scenario. $W_2$ seeks *first* to identify average ages, and *additionally* match up age peaks, so to indicate that it neglects this information is wrong. Nonetheless, we recognise that it could be confusing, and as such we follow the reviewer's suggestion to add an explanation for the behaviour:

> We reiterate that at a translation of 0 Ma, $W_2$ (and the KS distance) is still non-zero, reflecting the fact that even when the average ages are aligned, the shapes of the uni-modal and bi-modal distributions do not match. This illustrates the tendency of $W_2$ in geochronological data to prioritise aligning the average ages of distributions *before* considering matching individual peaks. Such behaviour contrasts with approaches that seek to only match probability peaks neglecting any information of absolute ages (e.g., Saylor et al. 2016).

**2. I maintain that the absolute distance along the x-axis does not encode geologically meaningful information absent some context (c.f., the geologically meaningful information encoded by KDEs that can be extracted without reference to the geological context and therefore can provide independent evaluation and verification of geological hypotheses (Sharman and Johnstone, 2017). See comment on line 168.**

We agree with the reviewer that 'unmixing' approaches such as that proposed in Sharman et al. (2017) can be used to extract source region distributions from detrital distributions, and that these do not depend on absolute distances along the x-axis. Indeed, in Section 3.1 we already propose that such methods are better suited to analyse mixtures samples than MDS using the Wasserstein distance. Consequently, it is not clear how we can further address this comment. Additionally, we emphasise this scenario is just one particular scenario in which geochronological distributional data is analysed, and in many other scenarios absolute distances along the x-axis provides useful geologic information (see Section 3.2 – 3.4).

**3. Due to the need to select a metric based on an expected outcome, as the authors suggest, I wonder if this whole enterprise does not descend into circular logic. In other words, the metric**

is chosen because we expect a certain conclusion and (lo and behold!) the method confirms that conclusion. Does this return the detrital geochronology/thermochronology back to the realm of subjectively assessing each distribution, even if we then represent that the subjective analysis that we conducted using "objective" metrics?

We recognise the concerns of the reviewer here to an extent, although feel that such concerns are beyond the scope of the manuscript. Further, for hypothesis driven research there will always be expected outcomes to be tested. As a result, an appropriate metric can be chosen on this basis. We agree that the scenario the reviewer proposes of repeatedly re-analysing a dataset with a variety of metrics and selecting the 'best' results is undesirable and would be analogous to 'p-hacking'. We have slightly modified our language in the Discussion to emphasise that the appropriate metric ought be chosen on the basis of the scientific question being answered, rather than on the dataset itself, which we agree could be confusing: *'the most appropriate dissimilarity metric to use will depend on the scientific question being answered.'*.

**4. Is there an internal check that would provide an assessment of whether the selected metric is successful? For example, for the data from Degraaf-Surpless et al. (2002), I calculated a stress of 0.14 for the MDS using the KS D value which is pretty high. Does the Wasserstein distance provide a better transformation (realizing that comparing between different metrics when conducting MDS is quite tricky)?**

Unfortunately we do not believe that such a simple check exists as what 'success' entails entirely depends on the scientific question being analysed, which will only partially depend on the dissimilarity metric chosen. We argue that 'stress' should not be considered as a measure of the success of MDS either. Stress is just the reflection of whether a dataset can be projected onto a two-dimensional plane without distortion. This is not the same as whether an MDS projection has actually recovered useful latent variables.

For example, consider a scenario where each sample is simply assigned a random number. Then, we calculate distances between samples by calculating the difference between these random numbers. MDS on these distances would produce a map with very low stress because the original values are distributed along a 1D line. However, whilst the map has low stress, it obviously has no actual inherent meaning.

However, we do recognise the importance of stress as an indicate of projcetion fidelity, so we have now added stress values to all of our MDS maps to the figure captions.

**5. If the method chosen is dependent on the specifics of the dataset and a prior analysis of the dataset, do the numerical metrics have any potential to illuminate (i.e., uncover latent features of) the dataset? See comment below on line 155.**

As discussed above, we do not advocate choosing a metric on the basis of datasets, rather the scientific *question* being analysed. For example, questions related to absolute geologic timings are well suited for analysis using the Wasserstein distance, whereas questions related on quantifying *amounts* of material are best solved using explicit mixing models.

**Minor Comments**

**Line 122: I recommend removing this discussion of linearity. This is really a function not only of the translation of the respective distributions, but also of the shape of the distributions themselves. In other words, translating a bimodal distribution past another bimodal distribution would produce non-linear results for the Wasserstein distance as well.**

[Figure]

Figure 1: **Translating bimodal distributions past eachother.** Note the linearity of $W_2$ with respect to offset, contrary to the suggestions of the reviewer.

The reviewer's statement '*translating a bimodal distribution past another bimodal distribution would produce non-linear results for the Wasserstein distance as well*' is factually incorrect. Translating two bimodal distributions past eachother, much like the example in Figure 2 result in a linear change in $W_2$ as shown in Figure 1 in this document. This linearity is to be expected from Equation 3, where as only the means of the distributions are changing, the change in $W_2$ will linearly depend on the change in the means (i.e., the offset). Consequently we do not wish to modify our original (correct) language here.

**Line 153: I think further caveats are warranted here. I don't think that the Wasserstein distance can be applied with certainty in all instances of identifying upsection trends in provenance changes. As one simple example, take the dataset below (from Smith et al. (2023)) where the Wasserstein distance fails to identify unimodal sample 1CCT3 as a unique population and instead lumps it in with the bimodal 1FCTC166. Counterintuitively the Wasserstein distance also places unimodes 2PCGT190 and 1DCGT243 closer to the bimodal cluster than other bimodal samples such as 1FCTC166, or even samples that share the same modes but in different proportions such as SJMT7. I realize that all of these metrics have their own caveats. I think the take-away for me is that the caveats need to be clear and that the basis for calculating the metric needs to be to be appropriate for the task (geochronology in this case). The authors can address the first of these issues by adding appropriate text to the**

**manuscript as it is. The second one forms the basis of my deeper concern as outlined in point 2 in the General comments.**

We agree that it would be unlikely the Wasserstein distance (or any metric) to automatically identify up-section provenance trends in all possible scenarios. However, we feel that this is an unreasonably high bar to pass, and certainly we do not believe that any dissimilarity metric already in use by the geochronology would pass such a test. Instead, as we discuss in Section 3.2, $W_2$ is *better* suited for this task than other metrics due to its sensitivity to temporal trends in source age distributions. We also agree with the reviewer that all metrics have caveats, and have attempted to detail those of $W_2$ in the revised Discussion in the manuscript. We have addressed point 2 above.

**Line 155: MDS of the cross-correlation coefficient provides something similar to the Wasserstein distance. There is less clustering between GV-42, -45, -40, and -64 than with the Wasserstein distance, but otherwise it is pretty close. I am not necessarily advocating that the authors present this approach. Rather, I am presenting alternative approaches to consider which may illuminate the data or a way forward.**

We agree with the reviewer that this MDS map reasonably identifies the upsection trend of these samples. However, it is perhaps surprising that it struggles to cluster the four unimodal samples given that cross-correlation is so designed to cluster samples with shared peaks. We hope that this manuscript provides a useful discussion for the geochronological community on how dissimilarity metrics ought be chosen given various advantages and disadvantages.

**Line 160, 170: What are the stresses associated with these MDS plots?**

We have now added 'Stress' values to the captions of all MDS plots.

**Line 168: I disagree that the absolute distance long the time-axis provides any useful information here. The interpretation that there is a break in exhumation between WBS7 and WBS8 would be the same if the older samples had modes at 100 Ma, instead of at 1,000 Ma. The relevant information is whether there is significant overlap in age distribution between the northern 4 samples and the southern 4 samples. Beyond this, the comparison between samples is not meaningful. (Obviously, it changes the geological interpretation whether the southern four samples have age modes at 100 Ma or 1,000 Ma, but that is not a function of the intersample comparison. The four southern samples reveal their own geological history without reference to any other samples.**

We disagree with the reviewer here as there is an additional temporal trend within the southern samples from the most southern sample (WBS1) to the most northern (WBS8). This ordering of the southern samples is identified by $W_2$ due to its sensitivity to the time-axis. The KS map however both fails to discover the ordering of the southern samples. Further, we note that the gap between the two clusters on the $W_2$ MDS map of is proportional to the (significant) relative age gap between the two exhumation signals. The KS map however (which cannot 'see' the different ages of the clusters) suggests that the difference between the two clusters of exhumations is smaller than the variability within each group. From a thermochronological perspective this might erroneously indicate that the different exhumation signals across the physiographic divide are quite similar, and that variations in exhumation history within each cluster are equally as important. So whilst the reviewer is correct that to simply extract two clusters, the KS distance is adequate (although we note that these clusters are poorly defined), but there is more geological information that can be meaningfully extracted from the data if $W_2$ is used.

**References**

Saylor, J. E. and K. E. Sundell (2016). "Quantifying comparison of large detrital geochronology data sets". *Geosphere* 12.1, pp. 203–220.

Sharman, G. R. and S. A. Johnstone (2017). "Sediment unmixing using detrital geochronology". *Earth and Planetary Science Letters* 477, pp. 183–194.